# Intracellular *Fusobacterium nucleatum* infection attenuates antitumor immunity in esophageal squamous cell carcinoma

Yiqiu Li [1], Shan Xing [2], Fangfang Chen[1], Qifan Li[1], Shuheng Dou [1], Yuying Huang[1], Jun An [3] ✉, Wanli Liu [2] ✉ & Ge Zhang [1] ✉

Currently, the influence of the tumor microbiome on the effectiveness of immunotherapy remains largely unknown. Intratumoural *Fusobacterium nucleatum* (Fn) functions as an oncogenic bacterium and can promote tumor progression in esophageal squamous cell carcinoma (ESCC). Our previous study revealed that Fn is a facultative intracellular bacterium and that its virulence factor Fn-Dps facilitates the intracellular survival of Fn. In this study, we find that Fn DNA is enriched in the nonresponder (NR) group among ESCC patients receiving PD-1 inhibitor and that the serum antibody level of Fn is significantly higher in the NR group than in the responder (R) group. In addition, Fn infection has an opposite impact on the efficacy of αPD-L1 treatment in animals. Mechanistically, we confirm that Fn can inhibit the proliferation and cytokine secretion of T cells and that Fn-Dps binds to the PD-L1 gene promoter activating transcription factor-3 (ATF3) to transcriptionally upregulate PD-L1 expression. Our results suggest that it may be an important therapeutic strategy to eradicate intratumoral Fn infection before initiating ESCC immunotherapies.

In recent years, immune checkpoint inhibitors (ICIs) have yielded unprecedented clinical benefits in multiple cancer types by restoring the function of antitumor T lymphocytes[1–3]. Nevertheless, ICI therapy fails to control neoplasia in a large number patients[3,4]. Esophageal squamous cell carcinoma (ESCC), the dominant histological type of esophageal cancer, is one of the most lethal malignancies and is highly prevalent in Asia[5]. Encouragingly, some clinical trials have shown the advantages of immunotherapy in ESCC. However, durable clinical responses are only observed in a fraction of patients[6,7]; thus, identifying the factors that influence the efficacy of immunotherapy and encouraging the application of individualized therapy are crucial for ESCC therapy.

Notably, the efficacy of ICIs and the gut microbiome have an inseparable relationship[2]. However, the influence of the tumor microbiome,

which is composed of tumor type-specific intracellular bacteria[8], on the effectiveness of immunotherapy remains largely unknown. Interestingly, *Fusobacterium nucleatum* (Fn), an obligate anaerobic bacterium, was identified as a facultative intracellular bacterium in our previous study[9]. Several studies have demonstrated that Fn functions as oncogenic bacteria and is implicated in the progression of ESCC[10–12]. Studies have illuminated that intratumoral Fn can promote tumor progression by generating a proinflammatory microenvironment[13], increasing the secretion of exosomes[14] or activating pathways such as NOD1/RIPK2/NF-κB[15]. Additionally, Fn adheres to and invades tumor cells via the virulence factor FadA and promotes carcinogenesis by activating the E-cadherin/beta-catenin pathway[16]. We recently confirmed that the virulence factor Fn-Dps can facilitate the survival of Fn and functions as an oncogenic factor to promote CRC metastasis[17].

---

[1]Department of Microbial and Biochemical Pharmacy, School of Pharmaceutical Sciences, Sun Yat-sen University, Guangzhou, China. [2]Department of Clinical Laboratory, State Key Laboratory of Oncology in South China, Collaborative Innovation Center for Cancer Medicine, Sun Yat-sen University Cancer Center, Guangzhou, China. [3]Department of Cardiothoracic Surgery, The Third Affiliated Hospital of Sun Yat-sen University, Yuedong Hospital, Guangzhou, China. ✉e-mail: anjun@mail.sysu.edu.cn; liuwl@sysucc.org.cn; zhangge@mail.sysu.edu.cn

Notably, Fn also promotes carcinogenesis by suppressing T-cell-mediated antitumor immunity. Higher numbers of intratumoural Fn have an inverse correlation with the numbers of T cells in CRC[18]. Intratumoural Fn is significantly and inversely associated with peritumoural lymphocytic reaction in ESCC[19,20]. Currently, very little is known about whether intratumoural bacteria influence the effectiveness of tumor immunotherapy. A recent study demonstrated that Fn can enhance the efficacy of PD-L1 blockade in CRC[21]. In contrast, several studies have indicated that enriched *Fusobacterium* is associated with shorter overall survival (OS) in anti-PD-1/PD-L1-treated lung patients[22,23].

In this work, we show that Fn is enriched in the nonresponder group among ESCC patients receiving immunotherapy. We further find that Fn can attenuate the efficacy of αPD-L1 treatment in ESCC animals. Mechanistically, the Fn virulence factor Fn-Dps can bind to the PD-L1 gene promoter activating transcription factor-3 (ATF3) to upregulate PD-L1 expression in ESCC cells. Fn can also inhibit the proliferation and cytokine secretion of T cells. Hence, we propose Fn might be a tool for personalizing the treatment of ESCC patients in the context of immunotherapies.

## Results

### F. nucleatum is enriched in the nonresponder group among ESCC patients receiving immunotherapy

First, we analyzed the differential expression of PD-L1 across TCGA cancer datasets. *PD-L1* expression was considerably higher in ESCA tumor tissues ($n = 162$) than in adjacent normal tissues ($n = 11$) ($P = 0.0011$) (Fig. 1a). The 5-year disease-free survival (DFS) of *PD-L1*-overexpressing patients was markedly lower than that of low *PD-L1*-expressing patients ($P = 0.002$) (Fig. 1b). Notably, Fn was detectable in both tumor tissues and adjacent normal tissues (Fig. 1c).

Next, we analyzed the correlation between PD-L1 and Fn in fresh ESCC samples. Both the relative abundances of Fn determined by qPCR analysis and the PD-L1 protein levels found by western blot analysis were significantly higher in tumor tissues than in adjacent normal tissues (Fig. 1d, e). The Fn DNA level was positively correlated with the PD-L1 protein level ($r = 0.769$, $P = 0.0035$) in ESCC tissues (Fig. 1f).

We then collected sera from 98 ESCC patients who received immunotherapy (Table S1). The progressive disease group (PD, $n = 20$) was defined as nonresponder (NR), whereas the partial response group (PR, $n = 46$) and stable disease group (SD, $n = 32$) were defined as responsive (R). ELISA showed that the PD group had higher serum anti-Fn IgG levels than the PR or SD group (Fig. 1g). We also analyzed the samples treated with anti-PD-1 alone (Fig. 1h). The data showed that the NR group ($n = 9$) had higher serum anti-Fn IgG levels than the R group (containing 8 PR and 5 SD). This result is consistent with the results shown in Fig. 1g. According to a receiver operating characteristic (ROC) curve analysis, the area under the curve (AUC) of serum anti-Fn antibody for PR to distinguish PD was 0.7516 (Fig. 1i).

Furthermore, 19 paraffin sections from ESCC patients (9 R + 10NR) were collected for the assessment of PD-L1 expression by immunohistochemistry (IHC) and Fn abundance by qPCR (table S2). We defined high expression of PD-L1 if the rate of positive cells was >45% and defined a high abundance of Fn if ΔCt (the cycle threshold) <10 (ΔCt = Ct(Fn)-Ct(18 S)). The rate of PD-L1-positive cells showed no differences between the R and NR groups, but the Fn abundance was significantly higher in the NR group than in the R group (Fig. 1j–k, S1–3) ($P = 0.0279$). Low protein levels of PD-L1 and low or no abundance of Fn DNA were more frequently found in the R group (Fig. 1l).

Taken together, these results indicate that Fn might exert opposite effects on immunotherapy with an anti-PD-1 antibody, although high levels of PD-L1 have been detected in ESCC patients.

### F. nucleatum infection has a detrimental impact on the efficacy of αPD-L1 therapy in ESCC preclinical models

To verify the above findings, male mice were preinfected with Fn by tail vein injection three times before AKR cell subcutaneous injection and then underwent αPD-L1 treatment (Fig. 2a). Immunofluorescence (IF) and qPCR analysis showed that Fn was present in the tumor tissues of Fn-infected mice (Fig. 2b, c). Both the tumor volumes and tumor weights of the noninfected mice undergoing αPD-L1 treatment were significantly smaller than those of the Fn-infected mice (Fig. 2d–f, S4a). H&E staining confirmed that the tumor masses were highly necrotic in the αPD-L1-treated group but not in the other three groups (Fig. 2g). No significant differences in body weight were found among the mice (Fig. S4b).

Obvious liver metastasis and lung inflammation were observed in the Fn-infected group, regardless of αPD-L1 treatment (Fig. S4c). Similar results were also observed in the group of female mice (Fig. S5). Next, to mimic patient treatments, the mice were infected with Fn and administered a regular dose of αPD-L1 once the tumor volume reached 250 ~ 350 $mm^3$ (Fig. S6a). As shown in Fig. S6b–f, the Fn+αPD-L1 group showed a smaller tumor volume than the Con and Fn groups, but still a significantly higher tumor volume than the αPD-L1 group, indicating Fn infection could decrease the effectiveness of αPD-L1 in larger-tumor-bearing mice, and the results are consistent with those shown in Fig. 2. Moreover, we observed a significant decrease in IFN-γ and TNF-α cytokine production in the serum of the Fn-infected group compared with that of the αPD-L1-treated group (Fig. S4d). To further explore the impact of Fn infection on the tumor immune microenvironment, we detected T-lymphocyte subsets in tumor tissues by flow cytometry. Fn significantly reduced the αPD-L1-mediated increase in $CD4^+CD3^+$ and $CD8a^+CD3^+$ cells and the activity ($GZMB^+$) of the infiltrated $CD8a^+$ T-cell population in the tumor region (Fig. 2h, S4e–h). Fn infection significantly increased the spleen weight (Fig. S4i).

Additionally, we established an AKR-esophageal cancer metastasis mouse model by tail vein injection (Fig. S4j). Fn infection significantly promoted lung metastasis in the metastasis model ($P < 0.01$) (Fig. S4k-l). Nevertheless, metastatic dissemination induced by Fn was not prevented by αPD-L1 treatment.

Collectively, our evidence demonstrates that Fn infection specifically jeopardizes the efficacy of anti-PD-L1 treatment in esophageal tumor-bearing mice. In particular, anti-PD-L1 treatment cannot inhibit metastasis in Fn-infected mice.

### F. nucleatum can enter into T cells and attenuate their activation

In order to investigate the suppressive effect of tumor growth by T cells in vivo, NSG mice were inoculated with E109 cells subcutaneously. Human $CD3^+$ T cells were infected with Fn (MOI 1:10) for 24 h as Fn-$CD3^+$ T cells (Fig. 3a, b, Fig. S7a). Once the average tumor size reached 250 $mm^3$, the mice were injected intravenously with human $CD3^+$ T cells or Fn-$CD3^+$ T cells. Compared with the uninfected group, Fn preinfection partly attenuated the inhibitory effect of tumor growth by T cells, and no difference in body weight was found among the three groups (Fig. 3c–f, S7b–d). Furthermore, $CD3^+$ T cells and Fn could be found in the tumor tissues of E109-bearing mice, as shown by IF analyses (Fig. 3g). We then investigated the effect of T cells combined with αPD-L1 treatment in Fn-infected NSG mice. Mice were infected with Fn by tail vein injection three times after subcutaneously injected with E109 cells. Then, mice were injected intravenously with human $CD3^+$ T cells and subjected to αPD-L1 treatment (Fig. 3h). As shown in Fig. 3i–k and Fig. S7e–f, both the tumor volumes and tumor weights of the noninfected mice undergoing αPD-L1 treatment were significantly smaller than those of the Fn-infected mice.

Because the largest immune organ and contains many lymphocytes and macrophages, we used a coculture system (splenocytes and tumor cells) to further investigate the role of Fn infection in affecting tumor immunity and tumorigenesis in vitro. As shown in Fig. 4a, ESCC cells were preinfected with Fn and then cocultured with splenocytes

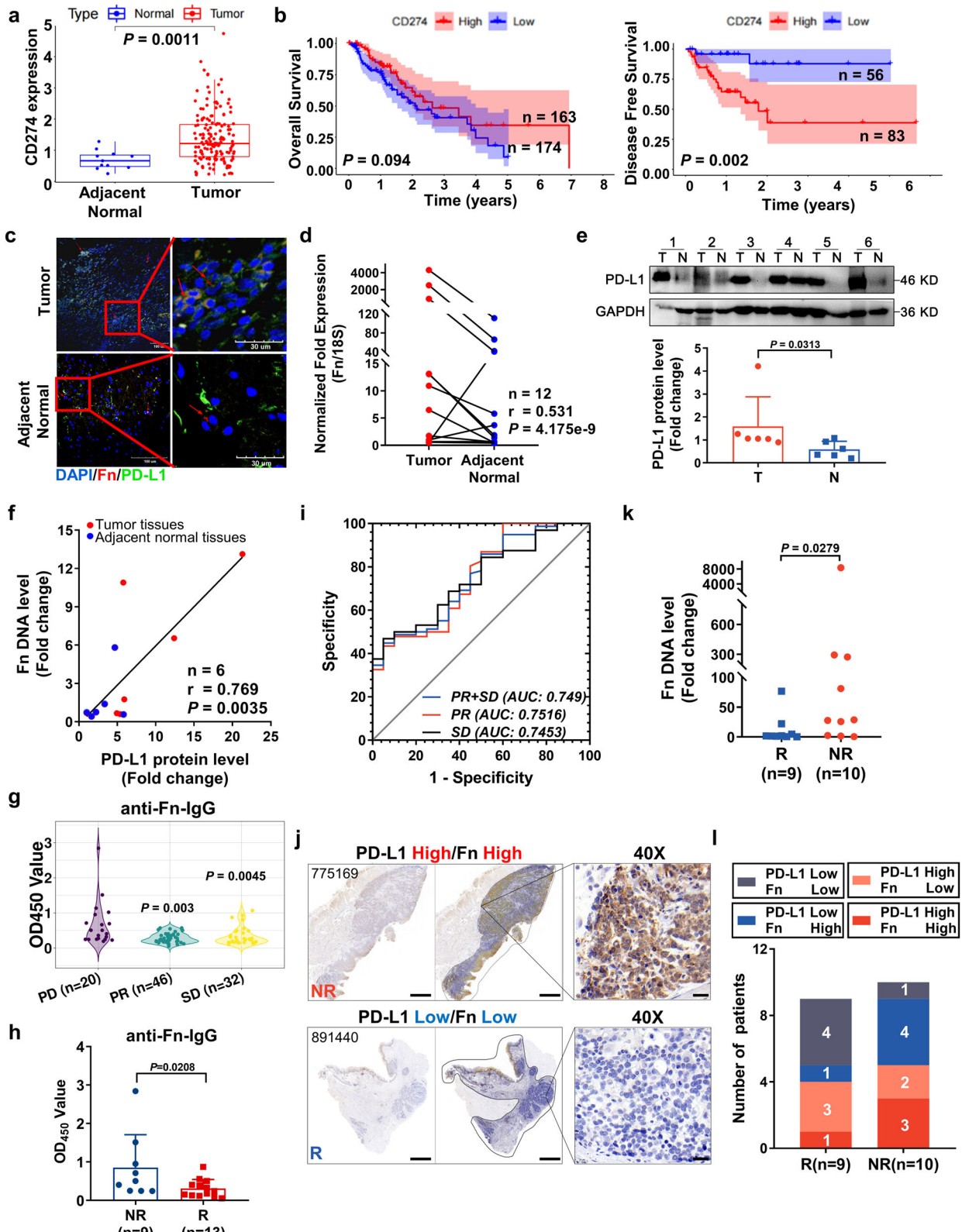

isolated from C57BL/6 mice. All three ESCC cell lines, E109, Kyse150 and AKR, exhibited significant necrosis and apoptosis after coculture with splenocytes for 60 h, as revealed by flow cytometry. Fn infection significantly attenuated the splenocyte killing efficacy (Fig. S8a). Furthermore, Fn infection could not rescue the cytotoxic effect from splenocytes, even with the use of a PD-1/PD-L1 blockade BMS202 (Fig. 4b, S8b).

In addition, CTL (CD8a+/CD3+ T cell) numbers were significantly decreased after infection with live Fn and killed Fn (K-Fn) for 48 h (Fig. 4c, d, S9a). To assess the killing effect of T cells with Fn infection, CD8+ T cells were infected with Fn at an MOI of 1:10, and we found that CD8+ T cells could not maintain their normal round morphology during Fn infection (Fig. 4e, S10). Fn entered CD3+ T and Jurkat cells after infection for 8 h (Fig. 4f). Both Fn and K-Fn could substantially

**Fig. 1 | F. nucleatum is enriched in the nonresponder (NR) group among ESCC patients receiving immunotherapy. a** Differential expression of CD274 between adjacent normal tissues (n = 11, blue) and tumor tissues (n = 174, red). Data were downloaded from http://xena.ucsc.edu/. Box and whisker plot; boxes depict the upper and lower quartiles of the data, and whiskers depict the range of the data. **b** Overall survival (OS) and disease-free survival (DFS) analysis of CD274 (mean ± SEM). **c** Dichromatic IF staining of Fn and PD-L1 in ESCC patient tissue sections. Scale bars: 100 μm (left) and 30 μm (right). **d** qPCR analysis of Fn in ESCC patient tumor and paired adjacent normal tissues (n = 12). **e** Immunoblotting analysis and quantification of PD-L1 in ESCC patient tumor (T) and paired adjacent normal (N) tissues (mean ± SD, n = 6). **f** Correlation analysis between the Fn DNA levels and PD-L1 protein levels. The samples are consistent with those in (**e**) (n = 6). **g, h** The OD450 values of anti-Fn-IgG in serum from ESCC patients were detected by ELISA (mean ± SD). Progressive disease (PD, n = 18); Partial response (PR, n = 49); Stable disease (SD, n = 34). P means vs. the PD group. Patients who were treated with anti-PD-1 alone (**h**). Non-responder (NR, n = 9) and responder (R, n = 13, containing 8 PR and 5 SD). **i** ROC curves for the diagnostic strength to identify total ESCC serum with indicators of PR + SD, PR and SD. **j** Images of IHC staining of PD-L1 expression in tumors from ESCC patients (n = 2 independent experiments with similar results). Scale bar: 2000 μm and 20 μm (40×). **k, l** qPCR analysis (**k**) of Fn in the R group (n = 9) and NR (n = 10) group and summary diagram (**l**). The statistical significance of results in **a**, **h** and **k** were determined by a two-tailed unpaired Mann–Whitney test. **b** was determined by Kaplan–Meier analysis. **d** and **f** were determined by a two-tailed nonparametric Spearman correlation analysis. **e** was determined by a paired nonparametric t-test. **g** was determined by a one-way ANOVA analysis.

decrease the production of IFN-γ and TNF-α by CD8+ T cells (Fig. 4g). Similar results were also observed in Jurkat cells (Fig. 4g). However, only active Fn exerted pro-proapoptotic effects on PBMCs, CD8+ T cells and Jurkat cells (Fig. 4h, S9b). Compared with the control and K-Fn treatments, live-Fn infection markedly inhibited the proliferation of Jurkat cells and increased the death of PBMCs and CD8+ T cells in a dose-independent manner, as shown by CFSE staining and proliferation analysis (Fig. 4i). Moreover, we assessed the ability of Fn to multiply inside Jurkat cells by a gentamycin protection assay. The survival and multiplication of intracellular Fn could not be maintained in Jurkat cells, the number of viable Fn showed a sharp decreasing trend (Fig. 4j), and the number of viable Fn per Jurkat cell also showed the same tendency (Fig. 4k). These results suggested that live Fn cannot survive inside T cells. In addition, LPS has been shown to reduce proliferation and induce apoptosis in Jurkat cells but generally exerts a weaker effect than Fn (Fig. S11).

Together, these data indicate that Fn could induce apoptosis and inhibit the proliferation and cytokine secretion of T cells. In addition, Fn-infected ESCC cells could protect against direct cytotoxicity from splenocytes in vitro and attenuate T-cell activation in vivo.

## Intracellular *F. nucleatum* survival upregulates the expression of PD-L1 in ESCC cells

We then observed that Fn could enter ESCC cells and localize in the cytoplasm by confocal assays and three-dimensional reconstruction of z-stack images (Fig. 5a, S12a). Moreover, we assessed the ability of Fn to multiply inside tumor cell lines by a gentamycin protection assay. The survival and multiplication of intracellular Fn could be maintained for 72 h, and active proliferation could be obtained in three host ESCC cell lines (Fig. 5b, c).

Next, flow cytometry analysis indicated that the expression of membrane PD-L1 was significantly increased in ESCC cells at 48 h postinfection with Fn (Fig. 5d, e, S12b). IF staining showed intracellular Fn and upregulated PD-L1 expression in Fn-infected cells (Fig. 5f, g, S12c). Additionally, both PD-L1 protein and mRNA levels increased gradually in a concentration-dependent and time-dependent manner after Fn infection, as revealed by qRT–PCR and western blot analyses (Fig. 5h–j). Notably, only live Fn infection upregulated the expression of PD-L1, whereas k-Fn, *E. coli DH5α* (Ec) or K-Ec did not affect PD-L1 expression (Fig. S12d, e).

Additionally, Fn infection upregulated the expression of PD-L1 in the tumor tissues of AKR-bearing mice, as shown by western blot and IF analyses (Fig. 5k, l). Altogether, these results indicate not only that Fn could survive and multiply inside ESCC cells but also that intracellular Fn could markedly upregulate PD-L1 expression in ESCC cells.

## Fn-Dps binds to the PD-L1 gene promoter ATF3 to transcriptionally upregulate PD-L1 expression

We further examined the expression of the virulence factor Fn-Dps by colabeling Fn-infected ESCC cell lines with specific anti-Fn and anti-Fn-Dps antibodies and observed the colocalization of Fn-Dps and Fn,

which were mainly distributed around the cell nucleus (Fig. 6a). The results showed that Fn, which produced Fn-Dps, was present across the cellular perimeter. Similar to the results observed with Fn infection, IF, western blot and flow cytometry analyses illustrated that Fn-Dps clearly induced the expression of PD-L1 in ESCC cells (Fig. 6b, c, S13). These data revealed that Fn can upregulate PD-L1 by producing Fn-Dps.

To further identify the upstream signaling pathway underlying Fn-Dps-mediated regulation of PD-L1 in ESCC cells, we performed two independent RNA sequencing analyses of Fn-infected and Fn-Dps-treated Kyse150 cells. Through transcription factor-binding site prediction, we found 5 overlapping transcription factors from 108 transcription factors for regulating PD-L1 expression (Fig. 6d, S14a). Among these 5 transcription factors, activating transcription factor 3 (ATF3) was identified as the top candidate (fold change = 3.2) (Fig. 6e, Table S4). Similar results were also observed from the analysis of the mRNA expression of these five genes in E109 and Kyse150 cells (Fig. 6f).

We further evaluated the correlation between ATF3 and PD-L1 based on TCGA datasets and found no significant difference in ATF3 expression between tumor tissues and normal tissues (Fig. S14b). Significantly lower overall survival (OS, P < 0.001) was observed in the high ATF3 expression group, whereas no significant difference in DFS was found in patients with EC tumors (Fig. S14c). Moreover, ATF3 gene expression was positively correlated with PD-L1 gene expression (P = 0.04, r = 0.16) (Fig. 6g). ATF3 reportedly regulates the PD-L1 levels; thus, we verified the correlation between ATF3 and PD-L1 in ESCC cells. We generated ATF3-overexpressing (ATF3-OE) ESCC cell lines, and a western blot analysis indicated that the PD-L1 levels were increased in ATF3-OE cells (Fig. 6h, S14d). Coimmunoprecipitation (Co-IP) revealed that ATF3 physically interacted with PD-L1 in E109 and Kyse150 cells (Fig. 6i). These results revealed that ATF3 could bind to and activate the PD-L1 promoter, resulting in PD-L1 transcription and consequently higher PD-L1 expression in EC.

Next, we chose siATF3-1 from three specific ATF3 siRNAs to silence ATF3 gene expression. Fn-Dps strongly upregulated PD-L1 expression in ESCC cells, whereas ATF3 knockdown suppressed this upregulation (Fig. 6j, S14e, f). Additionally, we verified the correlation between ATF3 and PD-1 in ESCC using a xenograft tumor model. Serial sections from the same tumor tissues were used for IHC analysis of ATF3 and PD-L1. The pathology results showed that the ATF3 levels were positively correlated with the PD-L1 levels (Fig. 6k). Double IF labeling for PD-L1 and ATF3 in mouse tumor tissues yielded similar results (Fig. 6l, S15). The Co-IP results also demonstrated the binding of Fn-Dps with ATF3 in E109 and Kyse150 cells (Fig. 6m). Additionally, we predicted potential ATF3-binding sites in the human CD274 gene promoter region using the JASPAR database. Two binding regions in ATF3 are shown in Fig. S14g, and promoter constructs containing mutations in these two regions were generated to induce ATF3-binding deficiency. Fn-Dps-treated cells had significantly higher PD-L1 expression than ATF3-Flag cells, whereas mutation of the CD274

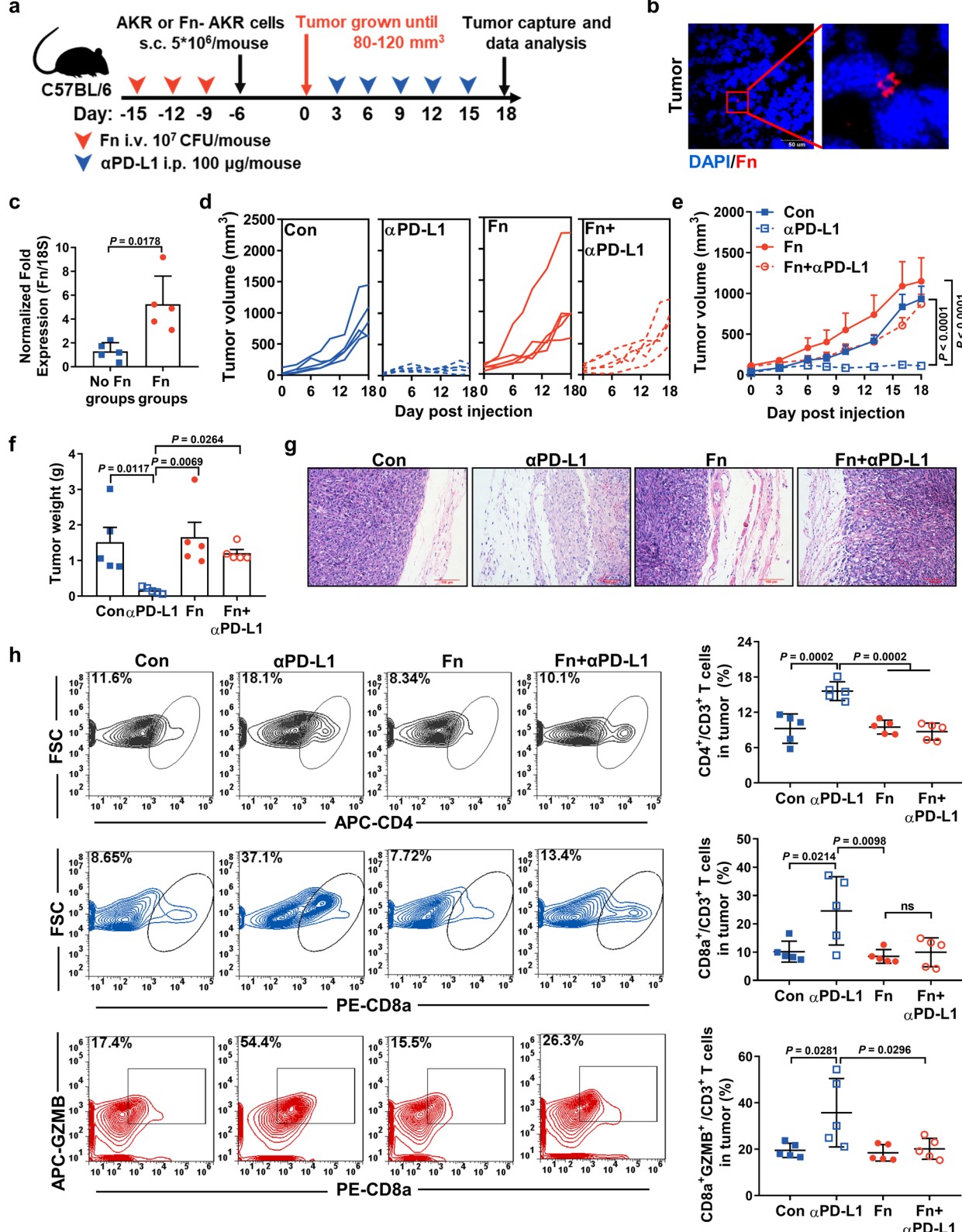

**Fig. 2 | *F. nucleatum* infection has a detrimental impact on the efficacy of αPD-L1 therapy in ESCC preclinical models. a** Schematic view of the administration plan. C57BL/6 mice (the experiment was done once; *n* = 5 per group) were implanted with 5 × 10⁶ AKR or Fn-AKR cells after Fn infection (*i.v.*, 10⁷ CFU/mouse) three times. Fn-AKR cells indicate that AKR cells were preinfected with Fn for 48 h. αPD-L1 (i.p., 100 µg/mouse) was administered once every 3 days. **b** IF staining of Fn in tumor tissue from the Fn group of C57BL/6 xenografts *n* = 3 independent experiments with similar results. Scale bar: 50 µm (left). **c** qPCR analysis of Fn in tumor tissues from the no-Fn and Fn groups C57BL/6 xenografts (mean ± SD, *n* = 5).

**d** Volume of each tumor (*n* = 5). **e** Mean tumor volume ± SEM (*n* = 5). **f** Mean tumor weight ± SEM (*n* = 5). **g** H&E staining analysis of tumors from C57BL/6 xenografts. Scale bar: 100 µm. **h** FACS quantification of CD4⁺CD3⁺, CD8a⁺CD3⁺ and GZMB⁺CD8a⁺ cells among CD3⁺ TILs in tumors from C57BL/6 xenografts (mean ± SD, *n* = 5). ns means not significant. The statistical significance of result in **c** was determined by a two-tailed unpaired Mann-Whitney test. **e** was determined by two-way ANOVA analysis for comparison at the endpoint. **f** and **h** were determined by a one-way ANOVA analysis.

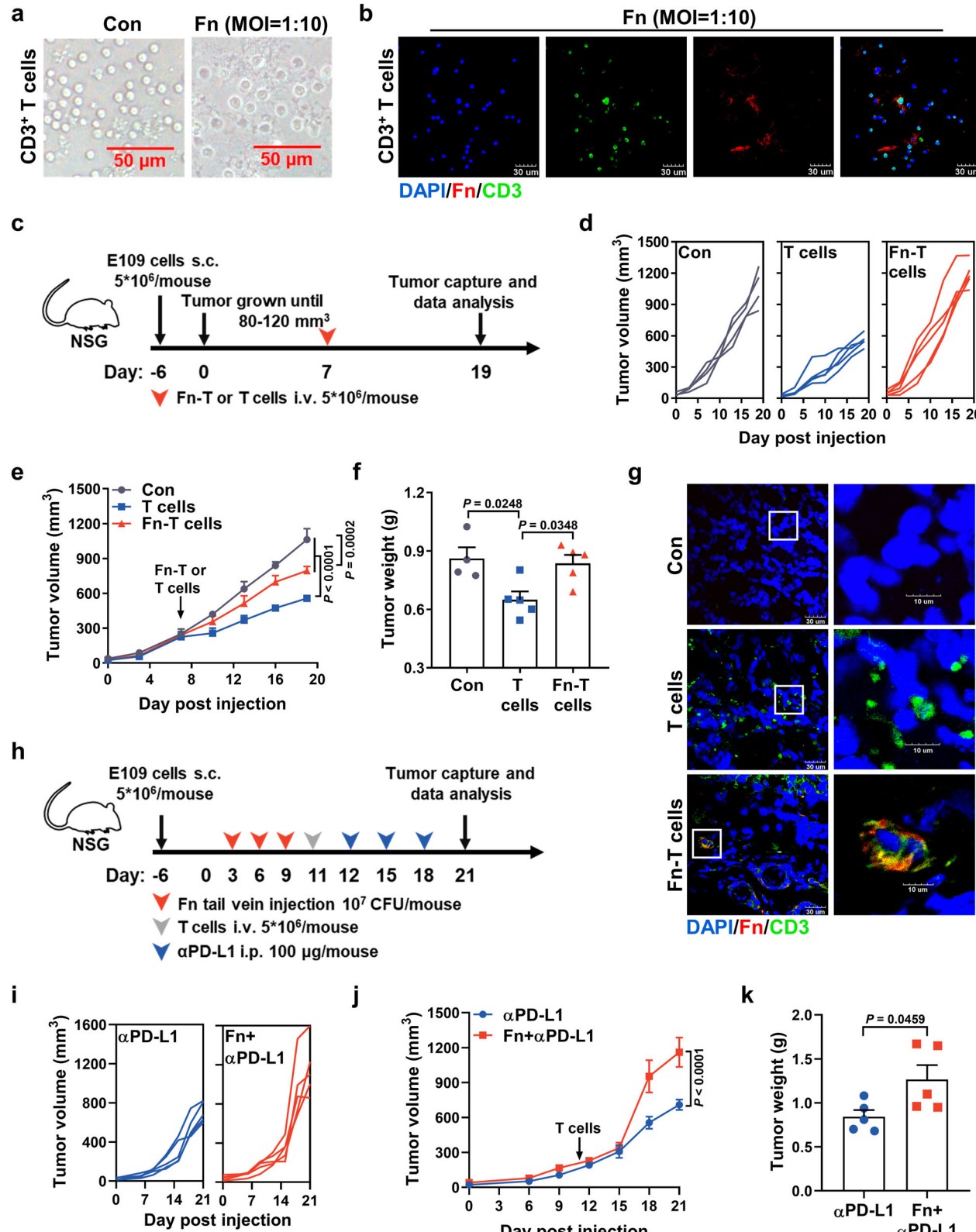

promoter reduced the expression of PD-L1 (Fig. 6n). We confirmed a similar tendency by Chromatin immunoprecipitation (ChIP)–qPCR (Fig. 6o), and the locations of the primers are indicated in P1 (Fig. S14h). Moreover, double IF analysis showed the colocalization of ATF3 with Fn-Dps inside ESCC cells (Fig. 6p). Western blot analysis showed a positive band of Fn-Dps in the nucleoprotein of ESCC cells (Fig. 6q). These results indicated that Fn-Dps could enter the cell nucleus to bind to ATF3 in ESCC cells.

Taken together, our results show that the virulence factor Fn-Dps could induce PD-L1 expression by upregulating its transcription through the activation of ATF3.

## Discussion

Here, we demonstrated that Fn upregulated PD-L1 expression in ESCC cells. Some bacteria that can enter and survive in host cells were found to upregulate surface PD-L1 in host cells. *Salmonella*

**Fig. 3 | *F. nucleatum* infection attenuats T-cell activation in vivo. a** Human CD3[+] T cells were obtained and purified from human peripheral blood mononuclear cells (PBMCs) and then infected by Fn for 24 h. Scale bar: 50 μm. **b** Dichromatic IF staining of CD3 and Fn in human CD3[+] T cells. The cells were infected with Fn (MOI of 1:10) for 24 h. Scale bar: 30 μm. **c–f** NSG mice (the experiment was done once; *n* = 4 mice in Con group and *n* = 5 mice in other groups) were implanted with $5 \times 10^6$ E109 cells, and the mice were then injected with $2 \times 10^6$ human CD3[+] T cells or Fn-CD3[+] T cells (Fn preinfection for 24 h) via the tail vein once the average tumor size reached ~250 mm³. Schematic view of the administration plan (**c**). Volume of each tumor (**d**). Mean tumor volume ± SEM (**e**). Mean tumor weight ± SEM (**f**). **g** IF staining of CD3 and Fn in tumor tissues from NSG xenografts. Scale bar: 30 μm (left) and 10 μm (right). **h, k** NSG mice (the experiment was done once; *n* = 5 mice per group) were implanted with $5 \times 10^6$ E109 cells and Fn infection (*i.v.*, $10^7$ CFU/mouse) three times. Then the mice were tail vein injected with $2 \times 10^6$ human CD3[+] T cells. αPD-L1 (i.p., 100 μg/mouse) once every 3 days. A schematic view of the administration plan (**h**). Volume of each tumor (**i**). Mean tumor volume ± SEM (**j**). Mean tumor weight ± SEM (**k**). Images in **a**, **b** and **g** were representative results of *n* = 3 independent experiments with similar results. The statistical significance in **e** and **j** were determined by two-way ANOVA analysis for comparison at the endpoint. **f** was determined by a one-way ANOVA analysis. **k** was determined by a two-tailed unpaired *t*-test.

reportedly induces PD-L1 expression in B cells and intestinal epithelial cells[24,25]. *Helicobacter pylori* (Hp) induces PD-L1 expression in gastric epithelial cells[26], and *Porphyromonas gingivalis* upregulates PD-L1 expression in prostate cancer cells[27]. It is well known that upregulated PD-L1 can help host cells escape attack by T cells. Of note, we also identified that Fn can survive and multiply in ESCC cells. We found that Fn infection upregulated PD-L1 expression in host cells to counteract immune attack and protect both intracellular Fn and host cells. These results suggest that Fn, as an intracellular bacterium, has developed a strategy to survive inside cells by hijacking tumor host cells.

Previous studies have found that Fn could secrete some virulence factors (such as Fada and Fap2) to facilitate growth, invasion and metastasis in the tumor microenvironment[16,28]. Our recent study identified Fn-Dps (a member of the DNA-binding ferritin-like protein family) as a virulence factor secreted by Fn-Dps is important in conferring tolerance to hydrogen peroxide and for the survival of bacteria[29,30]. As strict anaerobe, Fn is hypersensitive to oxidative stress, and we found Fn-Dps mutant cannot survive in isolation. We further investigated whether Fn-Dps could bind to ATF3 to upregulate PD-L1 expression. As an oncogene, ATF3 enhances proliferation and metastasis in a variety of tumors[31,32]. ATF3 was previously found to bind to the PD-L1 promoter and upregulate PD-L1 expression in melanoma and non-small cell lung cancer (NSCLC), which is consistent with our results[33]. Moreover, we identified that ATF3 can bind to the PD-L1 promoter in both human ESCC cell lines and C57BL/6 xenografts, although the ATF3-binding motif in the CD274 promoter differs among humans and mice. This is consistent with a study[33], they reported that ATF3 can bind to the PD-L1 promoter in human melanoma cell lines and B16F10 xenografts. These results suggested that the binding sites of ATF3 and PD-L1 are not unique. Together, our results indicate that Fn invades tumor cells and then releases Fn-Dps to enter the nucleus and bind to ATF3 to promote PD-L1 mRNA transcription. Upregulated surface PD-L1 further protects both tumor cells and their intracellular bacteria.

In the tumor microenvironment (TME), intratumoral bacteria can affect T cells, which play an important role in clearing pathogens and killing tumor cells. Our study demonstrated that both live and dead Fn induces apoptosis and necrosis of T cells. Early studies indicated that Fn immunosuppressive protein (FipA) is capable of impairing T-cell activation[34], and the outer membrane proteins Fap2 and RadD can induce human lymphocyte death[35]. Moreover, Fap2 can directly interact with TIGIT, leading to the inhibition of NK cell cytotoxicity and T-cell activities[36]. Our study is consistent with these conclusions and indicates that intratumoral Fn infection impairs T-cell function to facilitate the survival of both Fn and its host tumor cells. Thus, Fn has evolved mechanisms to evade host antimicrobial immune responses, leading to Fn-driven tumor progression.

Furthermore, we found that Fn infection has a detrimental impact on attenuating the efficacy of anti-PD-L1. We confirmed that Fn could enter T cells and suppress T-cell survival. Moreover, Fn infection reduced the αPD-L1-mediated increase in the CD4[+], CD8a[+] and GZMB[+] CD8a[+] T-cell populations in the tumor region. Similarly, a recent study

demonstrated that Hp could decrease the effectiveness of anti-PD-1 immunotherapies in NSCLC and gastric cancer[35,37]. Hp infection is significantly associated with increased expression of PD-L1[38]. Hp seropositivity is associated with reduced effectiveness of anti-PD-1 immunotherapy[39]. These results parallel our findings that among ESCC patients who had received immunotherapy, the anti-Fn antibody was significantly higher in the NR group than that in the R group. Moreover, we investigated whether Fn was more abundant in the TME in the NR group. The NR group tended to exhibit shorter survival than the R group, and this shorter survival was closely related to tumor metastasis. In this study, we found that Fn infection attenuates the antitumor immunity of PD-L1 blockers by impairing T cells and that Fn-infected ESCC cells could also protect against direct cytotoxicity from splenocytes in vitro and attenuate T-cell activation in vivo. The results of the tail vain model (Fig. S4j–l) showed obvious lung metastasis in the Fn and Fn+αPD-L1 groups. Based on these results, we hypothesized that metastatic dissemination induced by Fn could not be prevented by αPD-L1 treatment. Recent studies have demonstrated that Fn can expand myeloid-derived immune cells, which inhibit T-cell proliferation and induce T-cell apoptosis in CRC[40]. In addition, inoculation with Fn suppresses the accumulation of tumor-infiltrating T cells and promotes tumor progression, resulting in fewer CD4[+] and CD8[+] T cells in breast tumor-bearing mice[41]. These studies support our conclusion that Fn infection attenuates the antitumor immunity of PD-L1 blockers by impairing T cells.

Additionally, a previous study showed that upregulated PD-L1 expression in the TME can inhibit cytotoxic T lymphocyte (CTL) activation and thereby promote tumor immune escape[42]. PD-L1 expression has been reported to be associated with a worse prognosis in ESCC[43]. Previous studies have found that Fn infection could promote liver and lung metastasis in colon carcinoma-bearing mice[44,45]. Liver metastasis could promote antigen-specific T-cell apoptosis and an eventual attenuation in immune therapy efficacy[46]. These studies further support our conclusion that upregulated PD-L1 expression in Fn-infected ESCC cells is unfavorable for T cells.

We note a contrary conclusion with our study. A recent study reported that the Fn levels are correlated with improved therapeutic responses to PD-1 blockade in CRC patients and that Fn infection enhances the antitumor effects of PD-L1 blockade on CRC mice[21]. The difference in the scheme of the Fn-infected model is as follows: in the previous study, Fn was persistently intratumorally injected every 3 days after the tumor volume reached ~100 mm³, whereas in our study, we only injected Fn three times by tail vein before subcutaneous injection of tumor cells. In addition, previous studies have noted that intratumoral injection of certain bacteria may result in increased enrichment of tumor-specific antigens around tumors and cause oncolytic effects[47]. We believe that these differences would be an important cause of the obvious discrepancies among the results of animal experiments. As mentioned previously, Fn preinfection may lead to immunosuppressive activities by inhibiting T-cell responses, which results in the decrease in the effectiveness of αPD-L1 observed in our study. Moreover, previous studies have noted that high Fn levels are correlated with improved therapeutic responses to PD-1 blockade

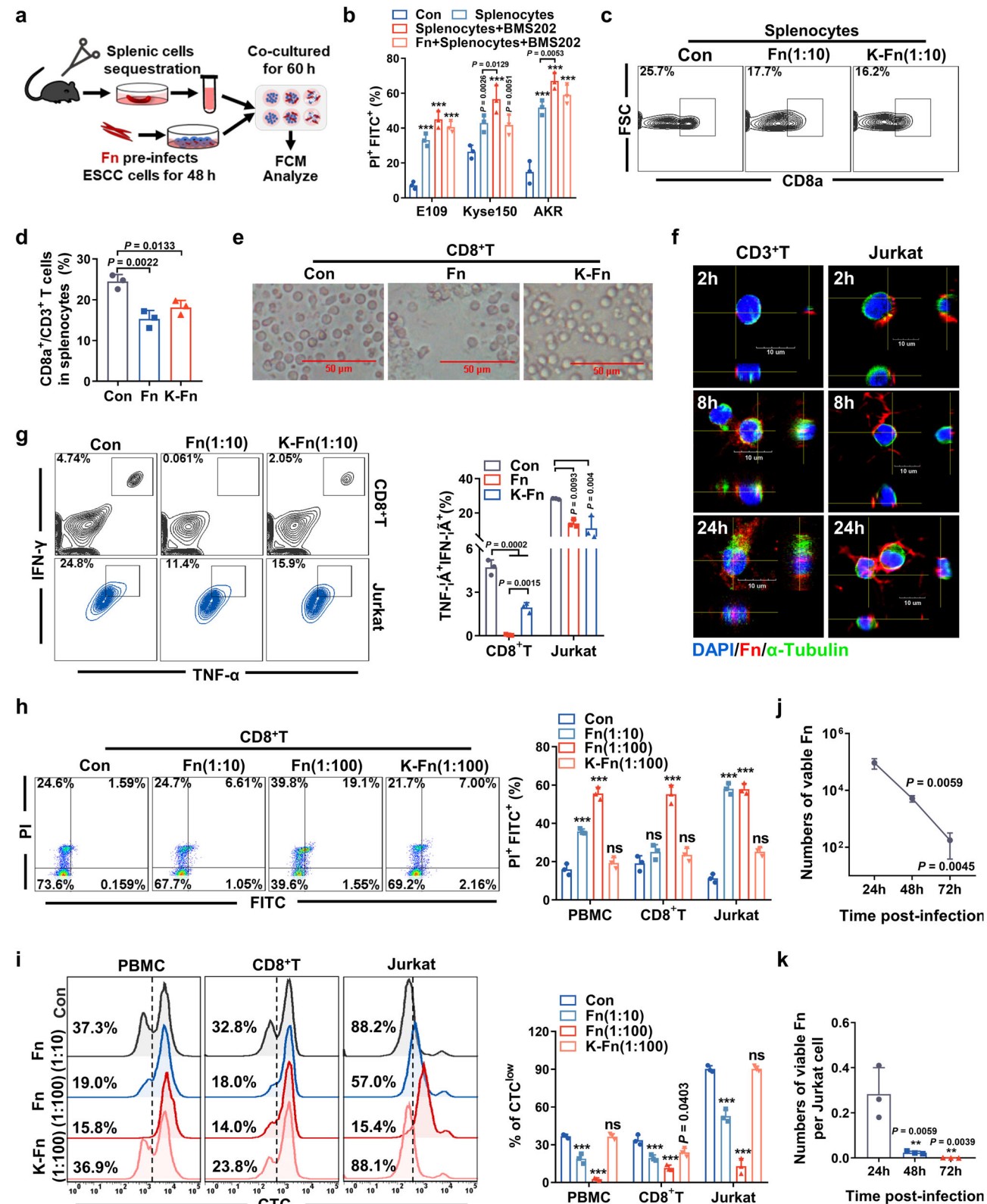

in CRC patients, whereas in our study, a higher abundance of Fn was found in the cancer tissues and serum from NR patients with ESCC. Interestingly, our previous study showed that Fn can upregulate the expression of indoleamine 2,3-dioxygenase in host cells, leading to tryptophan deficiency and impaired lymphocyte effector functions[9]. A recent study found that Fn could diminish sensitivity to anti-PD-1 mAb through its metabolite succinic acid in CRC[48]. A high abundance

of Fn is correlated with decreased efficacy of immunotherapy in CRC[48], and these results further support our findings. Moreover, two clinical studies noted that *Fusobacterium* is enriched in antibiotic-free NSCLC patients with an OS < 12 months compared with those with an OS > 12 months[22], and the abundance of Fn is enriched in NRs rather than responders among melanoma patients[8]. These studies also demonstrated that Fn could increase the accumulation of IFN-γ⁺CD8⁺

**Fig. 4 | F. nucleatum infection attenuats T-cell activation in vitro. a, b** Schematic diagram of the coculture study (A). Splenocytes were cocultured with Fn-infected ESCC cells (Fn preinfected ESCC cells for 48 h) for 60 h. Annexin V-FITC/PI-positive ESCC cells were counted by flow cytometry and quantified (**b**). The cells were pretreated with the PD-1/PD-L1 blockade agent BMS202 (1 mM) for 2 h before splenocytes were added. ***P < 0.0001 vs. the Con group. **c, d** FACS analysis of CD8a⁺ in CD3⁺ T cells and quantification. Splenocytes were infected with Fn or heat-killed Fn (MOI 1:10) for 48 h. **e** Representative images of CD8⁺ T cells infected with Fn or heat-killed Fn (MOI 1:100) for 48 h. Scale bar: 50 μM. **f** Three-dimensional visualization of α-tubulin and Fn in CD3⁺ T and Jurkat cells. Cells were infected with Fn (MOI of 1:10) for indicated hours. Scale bar: 10 μm. **g** FACS analysis and quantification of the intracellular cytokines TNF-α and IFN-γ in CD8⁺ T or Jurkat cells.

Cells were infected with Fn or heat-killed Fn (MOI 1:10) for 48 h. **h** Annexin V-FITC/PI-positive apoptotic cells infected with Fn (MOI 1:10 or 1:100) or heat-killed Fn (MOI 1:100) for 48 h. ***P < 0.0001 vs. the Con group. **i** The proliferation of PBMCs, CD8a⁺ T cells and Jurkat cells was analyzed by FACS. The cells were stained with cell trace CFSE (CTC) dilution first and then infected with Fn or K-Fn for 96 h. ***P < 0.0001 vs. the Con group. **j, k** Intracellular bacterial proliferation in Jurkat cells was assessed by a gentamycin protection assay. The numbers of total viable bacteria (**j**) and viable bacteria per cell (**k**) were determined by the serial dilution method. Images in **e** and **f** were representative results of n = 3 independent experiments with similar results. Results in **b–d, g–k** were presented as n = 3 biological replicates, mean ± SD, ns means not significant. The statistical significance of results in **b, d, g, h, i** and **k** were determined by one-way ANOVA analysis.

tumor-infiltrating lymphocytes, whereas a previous study noted that a decrease in the fold change in IFN-γ⁺CD8 T-cell anti-HIV Gag responses after anti-TIGIT and anti-PD-L1 blockade is correlated with higher *Fusobacteria* abundance[20]. Intratumoral bacteria were recently reported to modulate tumor progression and may have a negative impact on cancer immunotherapy[49]. Collectively, although the diversity and composition of bacteria differ among different types of cancer[50], studies have shown that Fn is prevalent in several types of cancer. These results indicate that a high abundance of Fn might be associated with poor outcomes in Fn-positive tumor patients treated with ICIs.

In summary, we showed that Fn infection partially blocks the activity of αPD-L1 therapy in murine tumor models. Mechanistically, as shown in Fig. 7, the virulence factor Fn-Dps, which is produced by Fn, upregulates surface PD-L1 in ESCC cells by binding to the transcription factor ATF3. The upregulation of PD-L1 protects tumor cells and intracellular Fn by evading T-cell attack. Furthermore, Fn could enter T cells and ultimately promote T-cell death, which would impair the tumor-killing ability of T cells. In addition, we found that the abundance of Fn is correlated with the effectiveness of cancer immunotherapies, which might be a tool for personalizing the treatment of ESCC patients in the context of immunotherapies. Our study suggests that Fn needs to be eradicated in immunotherapies.

## Methods

Ethics approval was granted by the Ethics Committee of the Third Affiliated Hospital of Sun Yat-Sen University, Yuedong Hospital (TAH-SYSU) (No. II2023-006-01) and Sun Yat-Sen University Cancer Center (SYSUCC) (No. B2022-677-01). All animal procedures were authorized by the SYSU Animal Experimentation Ethics Committee (SYSU-IACUC) and performed in accordance with the approved guidelines.

### Clinical samples

A total of 12 ESCC tissue samples and paired adjacent normal tissue samples were obtained from TAH-SYSU. A total of 98 ESCC serum samples and 19 ESCC paraffin sections were obtained from SYSUCC. All serum samples and paraffin sections were the samples remaining after clinical examination at TAH-SYSU or SYSUCC (June 2020 to March 2022). Tumor samples were collected and analyzed according to IRB-approved protocols. Fresh ESCC tissues were collected after surgical resection or biopsy, and serum samples were collected before the first αPD-1 treatment. All serum samples were separated by centrifugation at 1,000 g for 8 min at room temperature and then frozen at −80 °C until use. ESCC patients were defined based on Response Criteria in Solid Tumors (RECIST) 1.1. Based on magnetic resonance imaging (MRI) or computed tomography (CT) findings, their responses to treatment were evaluated as partial response (PR), stable disease (SD), or progressive disease (PD). Clinical response (R) was defined as PR and SD, whereas PD was defined as nonclinical response (NR). The background grouping of the study cohorts is shown in the online Supplemental Tables S1–S2.

### DNA extraction

DNA was extracted from sections of paraffin-embedded ESCC tissues using the TIANamp FFPE DNA Kit (TIANGEN, DP331, Beijing, China) according to the manufacturer's instructions.

### Strains and growth conditions

*Fusobacterium nucleatum* (ATCC 25586) was revived on blood agar plate (Huankai, Guangdong, China) for 48-72 h; Clostridium ventriculi (DSM 286) was revived on blood agar plate for 12 h; *Bifidobacterium bifidum* (ATCC 29521) was revived on de Man, Rogosa, Sharpe (*MRS*) agar plate for 48-72 h; *Bifidobacterium longum* (ATCC 15707) was revived on *MRS* agar plate for 24–36 h. *Parabacteroides distasonis* (ATCC 8503) was revived on blood agar plate for 36–48 h; *Akkermansia muciniphila* (ATCC BAA-835) was revived on blood agar plate for 48–72 h; *Porphyromonas gingivalis* (ATCC 33277) was revived on Brian Heart Infusion (*BHI*) agar plate with 10% defibrinated sheep blood and 0.5% vitamin K1 for 48 h. All bacterial were grown anaerobically by using AnaeroPack (Mitsubishi Gas Chemical Co., Japan) at 37 °C.

*Escherichia coli* (ATCC 25922) was revived on Luria–Bertani (*LB*) agar under aerobic conditions for 24 h at 37 °C.

### Fn-Dps protein expression and purification

Fn-Dps was obtained as we mentioned in our primary study[17]. In brief, Fn-Dps gene was amplified by polymerase chain reaction (PCR) and Fn (ATCC 25586) DNA as templates. The recombinant plasmids of pET28a-Fn-Dps were transformed into E. coli BL21 (DE3). The expression of the recombinant Fn-Dps was induced with 0.8 mM isopropyl β-d-1-thiogalactopyranoside (IPTG) for continuous 16 h cultivation at 25 °C. The endotoxin in Fn-Dps was removed using the Endotoxin Removal Kit (GenScript, Nanjing, China) according to the manufacturer's instructions.

### Cell culture and sample preparation

The human ESCC cell line Eca109 (E109) and Kyse150 (a kind gift from professor Musheng Zeng, SYSUCC, Guangzhou, China), Jurkat cell line (ATCC TIB-152), embryonic kidney 293 T cells (ATCC CRL-3216), and mouse ESCC cell line AKR (C945, Wuhan Sunncell Biotechnology, China) were cultured in RPMI 1640 or DMEM medium (Gibco, CA, USA), respectively. All cell lines used in this study are free of mycoplasma contamination and have been authenticated by STR profiling before the start of experiments.

Splenocytes were isolated from 6-week-old C57BL/6 mice and prepared into single-cell suspensions through a 70 μm cell strainer (Sorfa, Beijing, China) in DMEM (Gibco, CA, USA). Lymphocytes were obtained from human peripheral blood (PBMC) of healthy volunteer donors by density gradient centrifugation with lymphocyte separation medium (TBD, Tianjin, China). CD8⁺ T lymphocytes were purified using a CD8⁺ T-Cell Isolation Kit (130-096-495, Miltenyi Biotec, Germany) from single-cell isolates according to the manufacturer's instructions.

All cells were supplemented with 10% fetal bovine serum (FBS, Procell, Wuhan, China), 100 IU/mL penicillin and 100 μg/mL streptomycin, in a humidified atmosphere of 5% CO₂ at 37 °C.

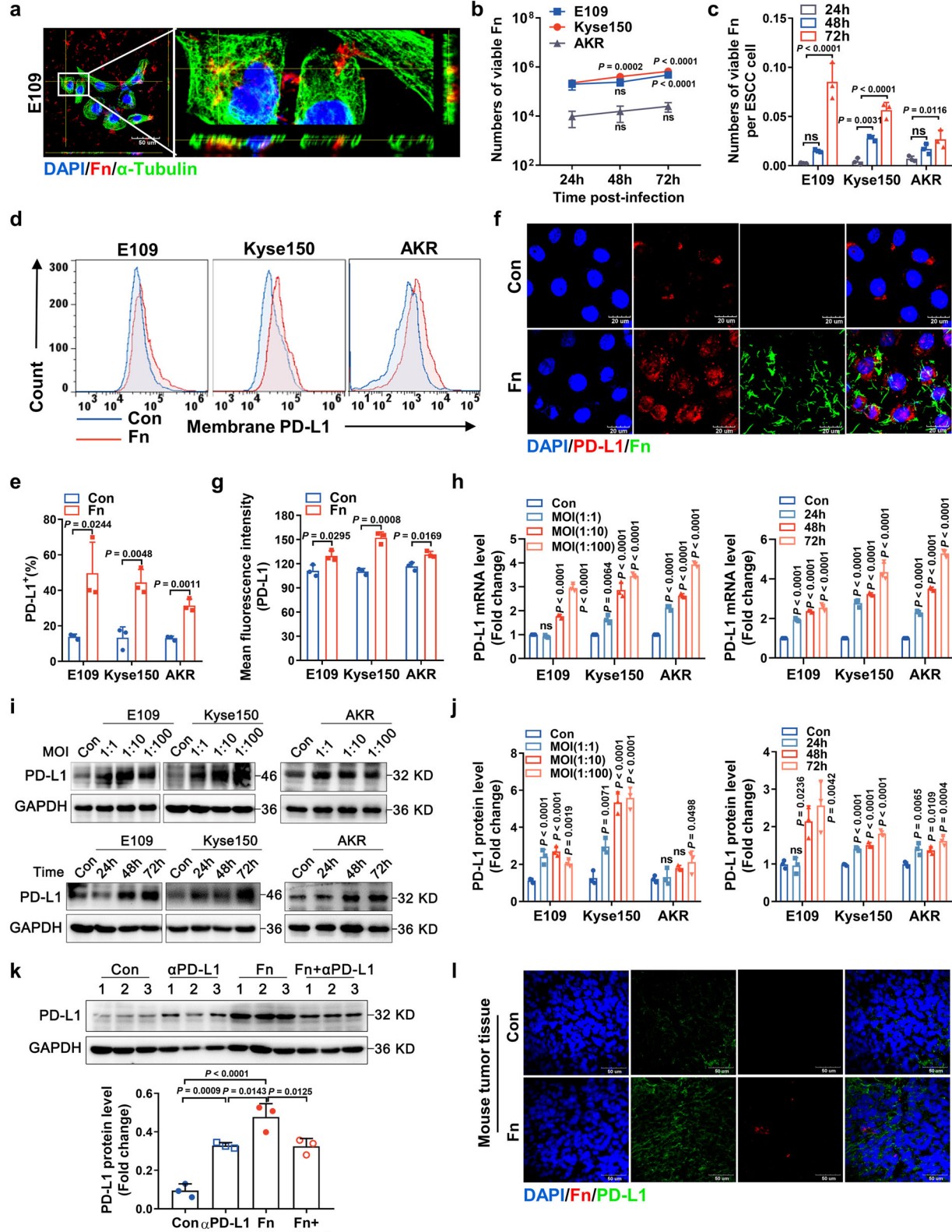

## Intracellular survival assays

Intracellular survival assays were performed as described previously[9]. ESCC cell lines or Jurkat cells were infected with live or heat-killed Fn (MOI 1:10) at the indicated time points. After infection, the cells were washed with PBS and then incubated for 2 h in RPMI 1640 or DMEM containing gentamicin (50 μg/mL) to remove extracellular Fn. Subsequently, the cells were lysed with 0.1% Triton X-100 and 0.01% SDS in PBS, and serial dilutions were spread onto Columbia blood agar plates (HKM, Guangzhou, China) to quantify the number of intracellular bacteria based on the mean of three replicate measurements.

## Splenocyte and tumor cell coculture study

ESCC cells were preinfected with Fn (MOI 1:10) for 48 h. Splenocytes (effector cells) were then cocultured with ESCC cells (target cells) for

**Fig. 5 | *F. nucleatum* can survive, proliferate and upregulate the expression of PD-L1 in ESCC cells. a** Three-dimensional visualization of α-tubulin and Fn in E109 cells. The cells were infected with Fn (MOI of 1:10) for 48 h. Scale bar: 50 μm (left). **b, c** Intracellular bacterial proliferation was assessed by a genta-mycin protection assay. E109, Kyse150 and AKR cells were lysed at the indicated time points after Fn (MOI of 1:10) infection, and the numbers of total viable bacteria (**b**) and viable bacteria per ESCC cell (**c**) were determined by the serial dilution method. *P* means *vs.* the 24 h group. **d, e** FACS of PD-L1$^+$ membrane expression in E109, Kyse150 and AKR cells after infection with Fn (MOI of 1:10) for 48 h and quantification. **f, g** Dichromatic IF staining of PD-L1 and Fn in E109, Kyse150 and AKR cells. The cells were infected with Fn (MOI of 1:10) for 48 h. **f** Representative images. Scale bar: 20 μm. **g** Quantification. **h** qRT–PCR analysis of PD-L1 in E109, Kyse150 and AKR cells after infection with Fn (MOI of 1:1, 1:10, 1:100) for 48 h or infection with Fn (MOI of 1:10) for 24, 48 or 72 h. *P* means *vs.* the Con group. **i, j** Immunoblotting analysis of PD-L1 in E109, Kyse150 and AKR cells after infection with Fn (**i**) and quantification (**j**). *P* means *vs.* the Con group. **k** Immunoblotting analysis and quantification of PD-L1 in tumors from C57BL/6 xenografts (mean ± SD; the experiment was done once; *n* = 3 mice per group). **l** IF staining of PD-L1 or Fn in tumor tissues from C57BL/6 xenografts. Scale bar: 50 μm. Images in a and l were repre-sentative results of *n* = 3 independent experiments with similar results. Results in **b, c, e, g, h** and **j** were presented as *n* = 3 biological replicates, mean ± SD, ns means not significant. The statistical significance of results in **b, c, h, j** and **k** were determined by one-way ANOVA analysis. **e** and **g** were determined by a two-tailed unpaired *t*-test.

60 h at a splenocyte:tumor cell ratio of 50:1. ESCC cells were pretreated with the PD-1/PD-L1 blockade agent BMS202 (Meilunbio, Dalian, China) for 2 h before the addition of splenocytes. The tumor cells were then washed with PBS to remove splenocytes, and the cells were resus-pended in Annexin-binding buffer and stained with Annexin V and propidium iodide (PI) for apoptotic cell analysis according to the manufacturer's instructions (Wanleibio, Shenyang, China).

### Flow cytometry analysis

For membrane PD-L1 expression analysis, ESCC cells were treated with Fn (MOI 1:10) or Fn-Dps (1 μM) for 48 h. The cells were then resus-pended in PBS and incubated with PE-CD274 or PE-isotype control for 40 min at 4 °C in the dark.

For apoptosis assessment, CD8$^+$ T lymphocytes were purified using a CD8$^+$ T-Cell Isolation Kit (130-096-495, Miltenyi Biotec, Ger-many) according to the manufacturer's instructions. Plate-bound anti-CD3 antibody was used to stimulate T cells. PBMCs, CD8$^+$ T cells and Jurkat cells were treated with Fn (MOI 1:10 or 1:100) or heat-killed Fn (MOI 1:100) for 48 h. The apoptotic cells were then analyzed as described above.

For mouse tumor or spleen samples, single-cell suspensions of C57BL/6-xenograft tumors were obtained by rapid and gentle strip-ping, physical grinding and filter filtration through a 70-μm cell strai-ner (Sorfa, Beijing, China). The cells were then stained with FITC-CD3, PE-CD8a and APC-CD4 for 40 min at 4 °C in the dark. After fixation and permeabilization with the True-Nuclear™ Transcription Factor Buffer Set (424401, BioLegend, CA, USA), the samples were stained with intracellular APC-GZMB antibody for another 40 min at 4 °C in the dark. Splenocytes were stained with FITC-CD3 and PE-CD8a for 40 min at 4 °C in the dark after infection with Fn or heat-killed Fn as described above.

The cells were washed twice with PBS to ensure that the excess dye was washed out. All the stained cells were then analyzed using a FACS Calibur cytometer (CytoFLEX S, Beckman, FL, USA). Data were collected from at least 10,000 cells per sample and further analyzed using Flow Jo 7.6 software.

The antibody details are listed in Table S3.

### Proliferation and cytokine secretion assays

The CellTrace™ Cell Proliferation Kit (C34554, Invitrogen, CA, USA) was used according to the manufacturer's instructions. Briefly, the cells were incubated with CellTrace™ CFSE (5 μM) for 20 min at 37 °C and then washed with FBS-supplemented PBS. After 5 days, the cells were harvested, and the dilution of CellTrace™ CFSE was evaluated with a FACS Calibur cytometer.

For the assessment of cytokine secretion, CD8$^+$ T cells were infected with Fn (MOI 1:10) for 48 h after stimulation with plate-bound anti-CD3 antibody. The cells were then stained with BV711-IFN-γ and PE-Cy™7-TNF-α for 40 min at 4 °C in the dark. Cytokine production was evaluated with a FACS Calibur cytometer.

The antibody details are listed in Table S3.

### Histology and immunofluorescence

For H&E staining, mouse tissue samples were fixed in 4% paraf-ormaldehyde, transferred routinely into paraffin, sectioned into 5-μm slices and stained with hematoxylin and eosin (H&E).

For IHC staining, antigens were retrieved by heating in citrate solution. Endogenous peroxidase was removed by adding 3% hydro-gen peroxide. The sections were incubated with the indicated primary antibody at 4 °C overnight. HRP-conjugated secondary antibody was then added, and the samples were incubated at room temperature for 2 h. The signal was developed for visualization with 3,30-diamino-benzidine tetrahydrochloride.

For double immunofluorescence (IF) staining, the cells were fixed, permeabilized and blocked using 4% paraformaldehyde (Beyotime Biotechnology, Shanghai, China) for 45 min at room temperature. Tissue sections were blocked for 30 min in PBS containing 3% BSA and 0.3% Triton X-100. The cells or sections were incubated with the indicated primary antibodies at 4 °C overnight. The secondary anti-body was then added, and the samples were incubated for 2 h. Nuclei were stained with 4,6-diamidino-2-phenylindole (DAPI, Fudebio, Hangzhou, China). After mounting, the cells were visualized using a laser scanning confocal microscope (FV3000, Olympus, Japan). The antibody details are listed in Table S4.

### siRNA

ATF3 siRNA (si-ATF3) and control siRNA (si-NC) were designed and synthesized by Sangon Biotech (Shanghai, China). Cells were seeded into 6-well plates at $2.5 \times 10^5$ cells/well and incubated overnight to reach 60–70% confluence. Transient transfections were then per-formed using Lipofectamine 3000 (Thermo Fisher, MA, USA). For luciferase reporter gene assays, 293 T cells were seeded at $1 \times 10^4$ cells per well in 96-well plates in DMEM (Gibco, Carlsbad, CA, USA) sup-plemented with 10% (v/v) fetal bovine serum (Procell, Wuhan, China) and cotransfected with 60 ng of each experimental plasmid (M35 empty vector, M35-ATF3, PL01-*CD274*, PL01-*CD274*-ATF3 mutation vector) per well. All plasmids were synthesized by GeneCopoeia (MD, USA). Renilla luciferase plasmid (40 ng per well) was used as an internal control. Twenty-four hours after transfection, the trans-fected cells were treated with Fn-Dps (1 μM) for another 48 h before the cells were harvested for luciferase assays. Luciferase activity was determined using the Dual-Luciferase Reporter Assay System (Glo-Max Navigator, WI, USA) according to the manufacturer's instruc-tions. The target sequences used for designing the siRNAs were as follows:

siATF3-1: 5'-CCUGAAGAAGAUGAAAGGAAATT-3';
siATF3-2: 5'-GCAUUUGAUAUACAUGCUCAATT-3';
siATF3-3: 5'-ACCUCUUUAUCCAACAGAUAATT-3';
siNC: 5'-UUCUCCGAACGUGUCACGUTT-3'.

### Polymerase Chain Reaction (PCR)

PCR reactions were performed in 50 μl reaction volume using Pri-meSTAR Max DNA Polymerase (TaKaRa Biotech, Dalian, China)

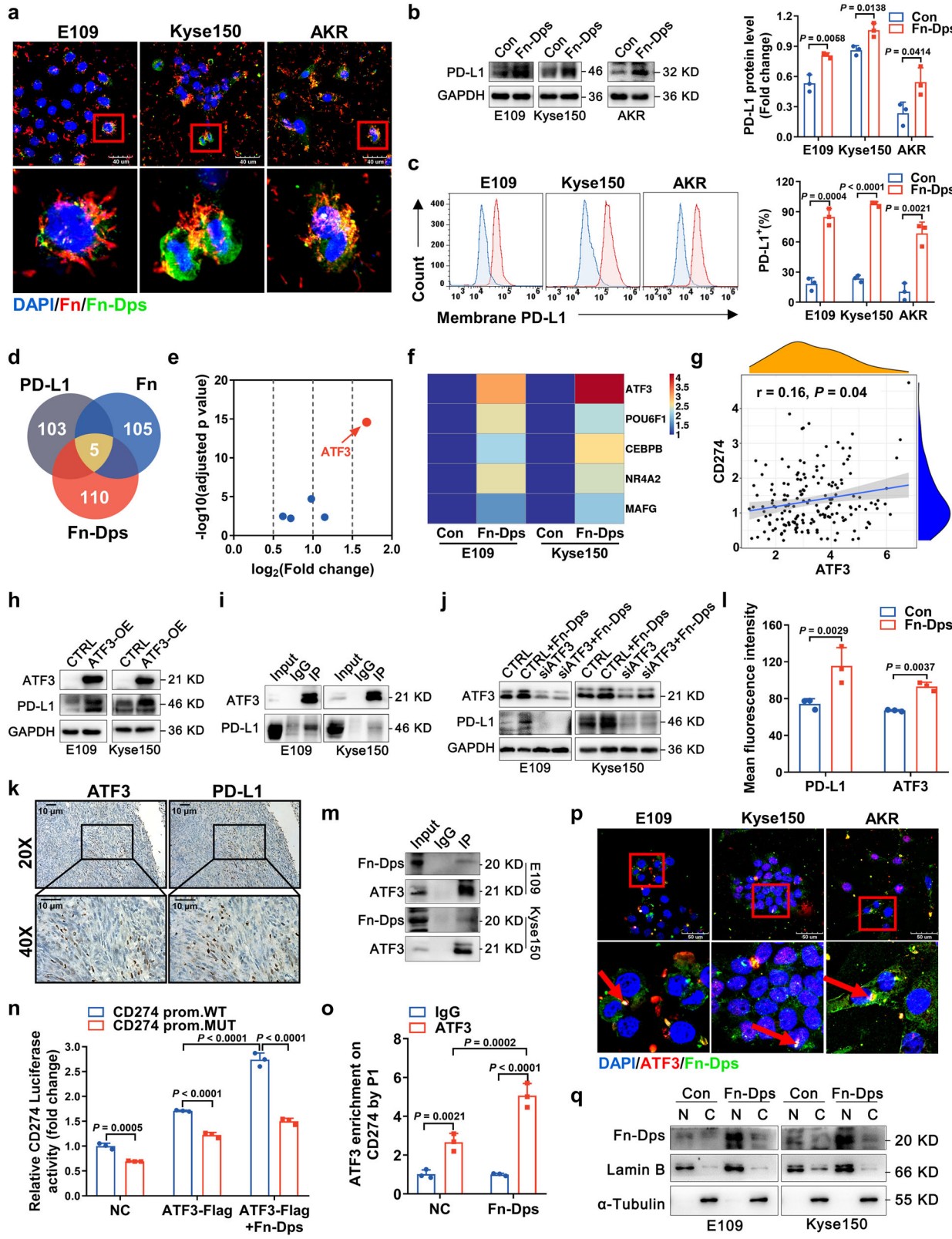

according to the manufacturer's instructions. PCR conditions for each amplicon included 95 °C for 5 min, 98 °C for 10 s, annealing temperature at 55 °C for 15 s and extension at 72 °C for 1 min. PCR amplification was visually confirmed using 2% agarose gel (TSINGKE, Nanjing, China). Agarose gel was imaged on a ChemiDocTM Imaging System (BIO-RAD, CA, USA).

**qRT-PCR**

Total RNA was extracted using AG RNAex Pro Reagent (Accurate Biology, Hunan, China) according to the manufacturer's protocol. cDNA was synthesized from total RNA using an Evo M-MLV Reverse Transcription Kit (Accurate Biology, Hunan, China). qRT–PCR was carried out on a BIO-RAD CFX96TM (BIO-RAD, Shanghai, China) using a

**Fig. 6 | Fn-Dps upregulates PD-L1 expression in ESCC cells by activating the transcription factor ATF3. a–c** Dichromatic IF staining of Fn and Fn-Dps (**a**), scale bar: 40 μm (top). Immunoblotting (**b**) and FACS (**c**) analysis of PD-L1. Cells were infected with Fn (MOI of 1:10) or treated with Fn-Dps (1 μM) for 48 h. **d–f** Venn diagram showing the unique and overlapping predicted transcription factors in different groups. The blue dots represent differentially expressed transcription factor genes (POU6F1, CEBPB, NR4A2 and MAFG). qRT–PCR analysis (**f**). **g** Analysis of the relevance of the CD274 (PD-L1) and ATF3 genes in ESCA tissues (TCGA database, *n* = 173). **h–j** ATF3 and PD-L1 expression was analyzed by western blotting. ESCC cells were transfected with ATF3 overexpression (OE) or negative control (CTRL) vectors for 48 h (**h**). Co-IP assay (**i**). Cells were treated with Fn-Dps (1 μM) for 48 h after transfection with siRNA against ATF3 for 48 h (**j**). **k, l** Representative images of IHC staining (**k**) and IF staining quantification (**l**) of ATF3 or PD-L1

expression in tumors from C57BL/6 xenografts. Scale bar: 10 μm. **m** The interaction of Fn-Dps with ATF3 in ESCC cells after treated with Fn-Dps (1 μM) for 48 h was detected by Co-IP assay. **n** Analysis of *CD274* WT or mutant promoter activity in 293 T cells transfected with ATF3-Flag and treated with Fn-Dps (1 μM) for 48 h. **o** ChIP–qPCR analysis of the relative enrichment of ATF3 at the *CD274* gene promoter in Kyse150 cells. **p, q** Dichromatic IF staining of ATF3 and Fn-Dps (p), scale bar: 50 μm (top). Immunoblotting of nuclear (N) and cytoplasmic (C) (**q**). Cells were treated with Fn-Dps (1 μM) for 48 h. Images in **a**, **h**, **j**, **m**, **p** and **q** were representative results of *n* = 3 independent experiments with similar results. Results in **b**, **c**, **n**, and **o** were presented as *n* = 3 biological replicates, mean ± SD. Statistical significance in **b**, **c** was determined by a two-tailed unpaired *t*-test. **l**, **n** and **o** were determined by two-way ANOVA multiple comparisons. **g** was determined by two-tailed nonparametric Spearman correlation analysis.

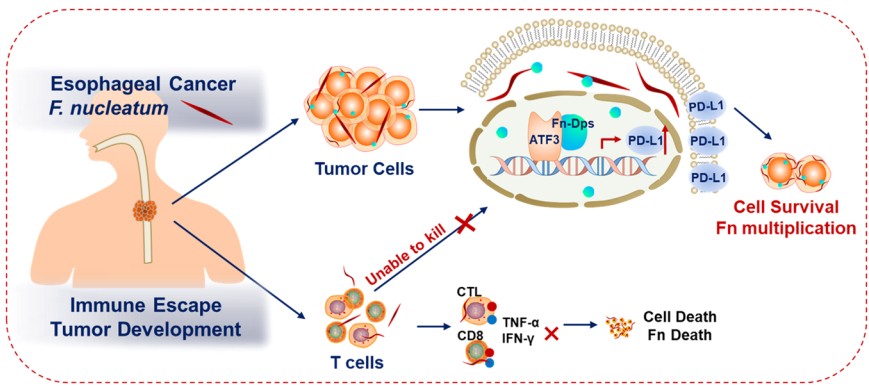

**Fig. 7 | Schematic illustration of the mechanism through which *F. nucleatum* promotes tumor immune escape in ESCC.** On the one hand, Fn can secrete virulence factor Fn-Dps, and then Fn-Dps combines with ATF3 to upregulate PD-L1 expression in tumor cells. On the other hand, Fn can also inhibit the proliferation

and cytokine secretion of T cells, Fn infection impairs T-cell function to affect the survival of both Fn and its host tumor cells. Finally, intracellular Fn infection promotes tumor immune escape in ESCC.

SYBR® Green Pro Taq HS qPCR Kit (Accurate Biology, Hunan, China). The sequences of the primers used in the qPCR analysis are listed in Table S5. Validation of Fn primer specificity is shown in Fig. S17.

## Western blotting (WB) and co-immunoprecipitation (Co-IP)
For western blotting, total cell proteins were separated by 10% SDS–PAGE and transferred to polyvinylidene difluoride membranes. The membranes were then incubated with primary antibody and horseradish peroxidase-conjugated secondary antibody. The proteins were detected using the ECL chemiluminescence system (NCM Biotech, Suzhou, China). Nuclear and cytosolic proteins were extracted using a Nuclear and Cytoplasmic Protein Extraction Kit according to the manufacturer's instructions (Beyotime, P0027, Shanghai, China). Lamin B and α-tubulin served as markers of the nuclear and cytoplasmic compartments. Image J software (Java 1.6.0_20, National Institutes of Health, USA) was used for the greyscale analysis of the scanned WB images. Validation of Fn and Fn-Dps antibodies specificity are shown in Fig. S16.

For Co-IP, to investigate the interaction between ATF3 and PD-L1 or Fn-Dps, the clarified supernatants were first incubated with anti-ATF3 antibody for 4 h at 4 °C. Protein A/G Magnetic Beads (Bimake, Shanghai, China) were then added, and after overnight incubation, the precipitates were washed five times with cell lysis buffer (Beyotime Biotechnology, Shanghai, China) and analyzed by western blotting. All antibody details are listed in Table S6.

## RNA sequencing analysis
RNA sequencing and analysis were performed by the Biomarker Technologies Corporation (Beijing, China). A total amount of 1 μg of RNA per sample was used as input material for the RNA sample

preparations. Sequencing libraries were generated using the NEBNext Ultra™ RNA Library Prep Kit for Illumina (NEB, USA) following the manufacturer's recommendations, and index codes were added to attribute sequences to each sample. Briefly, mRNA was purified from total RNA using poly-T oligo-attached magnetic beads. cDNA was then synthesized, amplified and purified. DESeq2 was used for differential expression analysis between sample groups to obtain differential expression gene sets between two biological conditions. Fold change ≥ 1.5 and *P* < 0.01 were used as screening criteria for the detection of differentially expressed genes. Transcription factor-binding sites (TFBSs) in the promoter regions of differentially expressed genes were predicted using the R package TFBS Tools. The reference for the transcription factor motif database is the JASPAR database (http://jaspar.genereg.net/). The raw sequencing data were uploaded to the GEO repository with the GEO accession number GSE222917.

## Transfection and luciferase assays
Transient transfections were performed using Lipofectamine 3000 (Thermo Fisher, MA, USA). For luciferase reporter gene assays, 293 T cells were seeded at $1 \times 10^4$ cells per well in 96-well plates in DMEM (Gibco, Carlsbad, CA, USA) supplemented with 10% (v/v) fetal bovine serum (Procell, Wuhan, China) and cotransfected with 60 ng of each experimental plasmid (M35 empty vector, M35-ATF3, PL01-*CD274*, PL01-*CD274*-ATF3 mutation vector) per well. All plasmids were synthesized by GeneCopoeia (MD, USA). Renilla luciferase plasmid (40 ng per well) was used as an internal control. Twenty-four hours after transfection, the transfected cells were treated with Fn-Dps (1 μM) for another 48 h before being harvested for luciferase assays. Luciferase activity was determined using the Dual-Luciferase Reporter

Assay System (GloMax Navigator, WI, USA) following the manufacturer's instructions.

## Chromatin immunoprecipitation (ChIP)-qPCR analysis

For ChIP–qPCR analysis, formaldehyde was added to the media, and the sample was incubated at room temperature for 10 min on a shaker. Subsequently, glycine was added and swirled gently to mix. The cells were washed three times with PBS and then lysed with 5 mL of Farnham lysis buffer (including protease inhibitors). The cells were separated in a centrifuge (5 min at 600 g) after scraping and transferred to conical tubes. The cell pellet was then rapidly frozen in liquid nitrogen for 10 sec. The frozen cell pellet was resuspended in 5 mL of Farnham lysis buffer (including protease inhibitors) and centrifuged for 5 min at 600 g. RIPA buffer containing protease inhibitors (1 mL) was added to the pellet before sonication after the supernatant was removed. The cell pellet was mixed with 25 μL of magnetic beads, and the supernatants were removed by the magnetic rack. Beads were washed three times with PBS and then incubated with 5 μg of antibody overnight. Afterward, the beads were washed three times before each sonicated sample was incubated with 50 μL of coupled antibody on a rotator overnight. Finally, the beads were incubated in a 65 °C water bath for 1 h, and the supernatant was collected in a centrifuge (3 min at 15,000 g). DNA fragments were purified using a GeneJET PCR Purification kit (Thermo Scientific, MA, USA). The purified DNA and specific primers for the CD274 promoter region were used to amplify the target DNA. ChIP-qPCR experiments were repeated three times. Kyse150 cells treated with vehicle or Fn-Dps (1 μM) for 48 h. Fold change indicates the indicated enrichment of the CD274 gene under the influence of Fn-Dps compared with IgG enrichment in cells treated with the vehicle control. The target sequences used in the ChIP assays were as follows:

 CD274 promoter-F: 5'- CTCTGACTTCCGTATTCCTC −3'
 CD274 promoter-R: 5'- ATGACTCACAGCCACTCT −3'

## ELISA

First, Fn protein antigens (4 μg/mL) were coated on ELISA plates at 4 °C overnight. The diluted serum samples (1:2000) were added to the coated wells and incubated for 1 h at 37 °C. Horseradish peroxidase-conjugated secondary antibody (Earthox, CA, USA) was then added, reacted at room temperature for 1 h and developed according to standard protocols.

Mouse blood samples were collected, and the plasma was separated by centrifugation at 860 g for 20 min. The plasma levels of TNF-α and IFN-γ were assessed using a mouse TNF-α or mouse IFN-γ ELISA Kit (MultiSciences, Hangzhou, China). The optical density (OD) at 450 nm was read using a microplate reader (iMark, BIO-RAD, Shanghai, China).

## Animal experiments

Four-week-old C57BL/6 mice and NOD.Cg-Prkdcscid IL2rgtm1Wjl/SzJ (NSG) mice were obtained from Nanjing Biomedical Research Institute of Nanjing University, China, and raised under pathogen-free conditions.

To evaluate the role of Fn infection in xenograft subcutaneous models of ESCC, C57BL/6 mice were inoculated subcutaneously with $5 \times 10^6$ AKR cells. After the tumors grew to 60–100 mm³ on average, the mice were randomly assigned to four groups ($n = 8$, containing 5 males and 3 females). Each group was treated as follows: (1) control group, i.p. injection of 100 μg of isotype control (*InVivo*MAb rat IgG2b isotype control, BE0090, Bio X Cell, NH, USA) and injection of 100 μL of PBS via the tail vein every 3 days; (2) αPD-L1 group, i.p. injection of 100 μg of αPD-L1 (*InVivo*MAb Anti-mouse PD-L1, BE0101, Bio X Cell, NH, USA) every 3 days; (3) Fn group, three injections of $10^7$ CFUs of Fn (100 μL) via the tail vein before tumor cell injection; and (4) Fn + αPD-L1 group, three injections of $10^7$ CFUs of Fn (volume of 100 μL) via the tail vein before tumor cell injection and i.p. injection of 100 μg of αPD-L1 every 3 days. A schematic view of the administration plan is shown in Fig. 2a. Similar treatments were administered in the AKR-esophageal cancer metastasis experiment (Fig. S4j) (No: SYSU-IACUC-2022-001239) and in larger tumor-bearing mice (Fig. S6a) (SYSU-IACUC-2023-001156).

To evaluate the role of Fn-Dps in xenograft subcutaneous models of ESCC, C57BL/6 mice were inoculated subcutaneously with $5 \times 10^6$ AKR cells. After the tumors grew to 60–100 mm³ on average, the mice were randomly assigned to two groups ($n = 3$). Each group was treated as follows: (1) control group, injection of 100 μL of PBS via the tail vein every 3 days; and (2) Fn-Dps group, injection of 5 μM Fn-Dps (100 μL) via the tail vein every 3 days (No: SYSU-IACUC-2022-000135).

To evaluate the suppressive effect of tumor growth by T cells in xenograft subcutaneous models of ESCC, NSG mice were implanted with $5 \times 10^6$ E109 cells. After the tumor grew to 250 mm³, the mice were randomly assigned to three groups. Each group was treated as follows: (1) control group, single injection of 100 μL of PBS via the tail vein; (2) CD3⁺ T group, single injection of 100 μL of PBS with $2 \times 10^6$ human CD3⁺ T cells via the tail vein; and (3) Fn-CD3⁺ T group, single injection of 100 μL of PBS with $2 \times 10^6$ human Fn-CD3⁺ T cells (Fn preinfection for 24 h, MOI = 1:10) via the tail vein. Plate-bound anti-CD3 (130-093-387, Miltenyi Biotec, Germany) antibody was used to stimulate CD3⁺ T cells. A schematic view of the administration plan is shown in Fig. 3c (No: SYSU-IACUC-2022-001834). To investigate the effect of T cells combined with αPD-L1 treatment in NSG mice. In Fn+αPD-L1, three injections of $10^7$ CFUs of Fn (100 μL) via the tail vein after tumor cell injection. Then, single injection of human CD3⁺ T cells via the tail vein and i.p. injection of 100 μg of αPD-L1 (*InVivo*MAb Anti-human PD-L1, BE0285, Bio X Cell, NH, USA) every 3 days. A schematic view of the administration plan is shown in Fig. 3h (No: SYSU-IACUC-2023-000922).

The tumor growth and body weight were monitored every 3 days. The tumor growth was measured in three dimensions twice a week using a caliper. The tumor volume was calculated using the following formula: (length × width²)/2. The maximal tumor size permitted by our ethics committee is 2500 mm³. The maximal tumor size in our animal experiments did not exceed the permitted maximal tumor size. All mice were sacrificed by dislocating the cervical vertebrae under anesthesia (1% sodium pentobarbital solution, 50 mg/kg).

## Statistical and reproducibility

The results are presented as the mean ± SD of three independent in vitro experiments at the cellular level. The tumor volume and tumor weight data from all mouse models are presented as the mean ± SEM. The data were examined to determine whether they were normally distributed using the Shapiro–Wilk normality test. For normally distributed data, independent sample *t*-tests were used for comparisons of two groups, and one-way ANOVA was used for comparisons among three or more groups. For data that were not normally distributed, the nonparametric Mann–Whitney test was used. Progression-free survival was calculated using a Kaplan–Meier survival curve. According to the cutoff value of CD274 (ATF3), the samples were divided into high-CD274-expression (ATF3-High) or low-CD274-expression (ATF3-Low) groups, and the cutoff value was generated by R programming language automatic analysis to minimize the *p*-value. The statistical analyses were conducted with GraphPad Prism7 (GraphPad Software, CA, USA). A *p*-value less than 0.05 was considered to indicate statistical significance.

## Reporting summary

Further information on research design is available in the Nature Portfolio Reporting Summary linked to this article.

## Data availability

All data generated or analyzed during this study are included either in this article or in the supplementary information files. The RNA

sequencing data have been uploaded to the Gene Expression Omnibus (GEO) repository, the accession number is GSE222917. All the TCGA data can be downloaded from http://xena.ucsc.edu/(GDC TCGA DATA SETS), and the analysis of TCGA data were implemented in R programming language (version: 4.1.3). Genome browser display of ATF3-binding events on the promoter and body of CD274 (PD-L1) in HCT116 cells (Fig. S14h), data can be found in http://cistrome.org/db/#/. ATF3-binding sites in the human CD274 gene promoter region was predicted using the JASPAR database (http://jaspar.genereg.net/). Source data are provided with this paper.

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

## Acknowledgements

This work was supported by the National Natural Science Foundation of China (No. 81972289 (to G.Z.)) and the Natural Science Foundation of Guangdong Province (No.2019A1515010500 (to G.Z.)).

## Author contributions

Y.L., W.L., and G.Z. designed the project. Y.L., F.C., Q.L., S.D., and Y.H. performed the experiments. S.X., J.A., and W.L. collected the clinical samples. Y.L., F.C., and S.D. analyzed the interpreted the data. Y.L. and G.Z. wrote the manuscript.

## Competing interests

The authors declare no competing interests.
