## [Peer Review File · Nature Communications]

Intracellular *Fusobacterium nucleatum* infection attenuates antitumor immunity in esophageal squamous cell carcinomaEditorial Note: Parts of this Peer Review File have been redacted as indicated to remove third-party material where no permission to publish could be obtained.

REVIEWER COMMENTS

Reviewer #1 (Remarks to the Author):

The main findings of this study are that high abundance of *Fusobacterium nucleatum* (Fn), and high anti-Fn antibody titer correlate with poor response to anti-PD-1 treatment in ESCC. The authors reveal that Fn infection upregulates PD-L1 expression in host cells to counteract immune-cell attack and protect both the host tumor cell and intracellular Fn. Their results demonstrate that the Fn virulence factor Fn-Dps enters the nucleus of the host cell, binds the ATF3 transcription factor that then attaches an identified motif in the promoter of the PD-L1 gene to upregulate PD-L1 expression.

The results are persuasive, and of interest both mechanistically and clinically.

Major Points

The findings of this report agree with a previous one on colorectal cancer (Yaohui Gao et al, Signal Transduction and Targeted Therapy 2021) that found that Fn can induce PD-L1 expression. However, in the previous report high Fn levels were found to correlate with improved therapeutic responses to PD-1 blockade. The authors discuss this disagreement in the discussion, but not in sufficient depth.

The authors demonstrate that Fn infection upregulates PD-L1 expression, however the immunotherapy used (Table S1) was anti-PD1. Should increase of PD-L1 have a dramatic effect?

The study reveals a correlation between high levels of intratumoral Fn and failure in response to anti-PD-1 treatment in ESCC. This suggests that anti-PD-1 treatment may be futile in ESCC colonized by Fn. To resolve this point, it would be advantageous to conduct a TCGA dataset screening of all ESCC patients with Fn and compare the survival rates of patients who received anti-PD-1 treatment versus those who did not. This analysis would offer valuable insights whether immunotherapy for patients with ESCC that is colonized by Fn is beneficial or not.

Use of an Fn mutant in Dps as a control, would greatly strengthen the conclusions of this study.

Minor points

The report is concise and informative. However, in general, the description in the methods is short and insufficient. This is also true for the Figure legends, which makes following the presented results rather laborious.

In addition, insufficient details are supplied for the statistical analysis: "GraphPad Prism7 (GraphPad Software, CA, USA). The results are the means \pm standard deviation (SD) of three independent experiments. A p-value of less than 0.05 was considered statistically significant". What were the tests used for each comparison? Where corrections made for multiple comparisons with the same data?

It is stated that "The results are the means \pm standard deviation (SD) of three independent experiments." Please confirm that this ("of three independent experiments") is true for each displayed result.

Furthermore, Median rather than Mean seems more fitting for most of the displayed results that do not appear to be normally distributed.

Add the "n" value, in the legend of each panel of each figure.

All bacteria names (including in references) should appear in "italics" style.

figs 1 b-c: What is the cutoff separating CD274 high from CD274 low? Please present an additional

corresponding color-coded (red, blue) graph with all of the points (values) of the CD274 high and low tumors.

Data verifying the specificity of the homemade antibodies used for identifying Fn in the tissues, and the antibodies used to detect Dps is not provided.

In Fig S3 it is stated that "C57BL/6 mice (5 male per group) were implanted with 5×10^6 AKR cells BEFORE Fn infection (i.v., 107 CFU/mouse) three times." In the methods and in Fig 2A it is stated that "C57BL/6 mice (5 male per group) were implanted with 5×10^6 AKR cells AFTER Fn infection (i.v., 107 647 CFU/mouse) three times." Which is the correct description for the experiment?

In general, the protocols lack details and hard to reproduce. For example, in lines 448-450: "For double immunofluorescence (IF) staining, the cells were fixed, permeabilized and blocked using 4 % paraformaldehyde (Beyotime Biotechnology, Shanghai, China) according to the manufacturer's protocol.". I find it hard to follow which manufacturer is referred to?

Line 146: "(P < 0.05)" can be deleted.

Lines 96-97 and Fig. 1c: "Notably, Fn was detectable in both tumor tissues and adjacent normal tissues (Fig.1C)." I can't find the detailed description how this was performed. Furthermore, why was Fn quantified in the RNA level (RT-qPCR) and not in the DNA level (qPCR) (as I understand was performed for Fig 1F), which is more reliable for bacterial quantification.

Please supply proof of the specificity of the primers use, demonstrate that a single band, with the expected size and sequence is obtained using each of these primers sets.

Fig. 1F: What do the blue and red colors represent, and what is "Fold Change"?

Fig. 1K: Is "Fold change" the correct title in the y axis?

In Fig S6, is FITC representing the tumor cells (please indicate in the Fig legend)? Please supply the statistical analysis for Fig S6b.

Lines 164-165: rephrase: "Furthermore, Fn infection could not rescue the cytotoxic effect from splenocytes after treatment with the PD-1/PD-L1 blockade BMS202".

Lines 177-178: Where is the data supporting the conclusion that "Of note, live Fn culture from the infected cells revealed that Fn cannot survive inside T cells"? I would also restrict this to "under the tested conditions, live Fn culture from the infected cells revealed that Fn cannot survive inside T cells". Please also show a control demonstrating that the fusobacteria were killed by the T cells, and not by the exposure to oxygen.

Lines 191-193 and Fig 4 B-C: is the increase in the numbers of intracellular fusobacteria statistically significant?

Line 499: What is IP in "washed five times with IP and...".

Lines 249-250, and legend to Fig 5N: I might have missed it, but I fail to find the discretion of the mutagenesis performed in the CD274 gene promoter region (including the description and the nature of the mutations).

Is the ATF3 binding motif in the CD274 promoter conserved both in human and in mouse (explaining how Fn-Dps can operate in both species)?

Line 296: Define "NSCLC".

312-314: Please rephrase "Liver metastasis could promote antigen-specific T-cell apoptosis and

eventual immune therapy efficacy diminishment”.

Line 316: Delete “mentioned”.

341-343: Rephrase “by binding to the PD-L1 gene promoter ATF3.”

Carefully go over the English style in the “In summary” paragraph ending the discussion.

Reviewer #3 (Remarks to the Author):

In this study, the authors investigated the presence of *Fusobacterium nucleatum* (Fn) in esophageal squamous cell cancer (ESCC) patients undergoing anti-PD1 therapies. Higher levels of Fn DNA were detected in ESCC patients non-responsive to anti-PD1 therapies. Using a mouse model with subcutaneous injection of ESCC cells, the authors showed that Fn infection was altering negatively the efficacy of anti-PD-L1 treatment. Mechanistic studies using ESCC cells and/or T cells revealed that Fn inhibits the secretion of cytokines and the proliferation of T cells while regulating the expression of ATF3 and PD-L1. Although there is some novelty to these findings, the role of Fn in ESCC and/or the response to PD-1/PD-L1 therapies have been reported before, such as in lung cancer. Furthermore, some important concerns need to be addressed.

Major concerns:

- While the mouse findings are interesting, the findings would be strengthened if this could be repeated in a second ESCC line. Or by using the 4-NQO model if in need of an immune competent model
- The experimental design for the mouse experiments in Figure 2 seems inadequate. For in order to mimic patient treatments, it would have been better to let the tumor grow to a certain size and then to perform infections and PD-L1 treatment.
- The use of certain ESCC cell lines in some experiments and other ones in different experiments is very confusing. It appears like the authors picked the cell line that would give them the result they wanted. Performing each experiment in at least 2 cell lines would be better
- For Figure 5, the rationale for ATF3 and the data presented is weak. The correlation between ATF3 and PD-L1 is low.
- Important details in the method section are lacking for the patient cohort undergoing anti-PD-1 treatment. First of all, it appears that some patients underwent chemotherapy while others did not. Were these cofounders taken into consideration? Second of all, the criteria for patients to be classified as responders vs non-responders were not mentioned.

Other concerns:

- This manuscript would really benefit from professional English editing services
- In the results, mention that analyzed Fn in ESCA tissue? Is that a typo or was it a combination of ESCC and EAC? This should be clarified
- Figure 1 A, C: normal should be labeled adjacent normal. Same for methods and results. This is misleading to call them normal.
- Figure 1D: says n=12 but in the text the authors mention that they collected 6 specimens. Where is the n=12 coming from? Counting 6 ESCC and 6 adjacent normal is wrong and misleading
- Figure 1E, given that GAPDH is uneven, densitometry should be performed. Same for all other WB included.
- How do the authors reconcile that they observed no change in metastasis in the xenograft model, but that they did in the tail vein model?
- Figure 2D should say volume not volume
- To strengthen the rationale for the experiments in Figure 3, the authors should show a staining of a positive Fn and CD3 co-staining in ESCC
- Figure 4L does not show that much expression of Fn. Looking at Figure, it appears that Fn levels are only going up 10 folds compared to the 4000 fold in human. ESCC. Furthermore, quantitative experiments of Fn and PD-L1, such as WB and WB would be more convincing.

Reviewer #4 (Remarks to the Author):

The authors examined intratumoral *Fusobacterium nucleatum* in relation to the effectiveness of immunotherapy in esophageal squamous cell carcinoma (ESCC), and found that Fn can inhibit proliferation and cytokines secretion of T cell, and Fn-Dps binds to the PD-L1 gene promoter activating transcription factor-3 (ATF3) to transcriptionally upregulate PD-L1 expression. This paper is very interesting. However, there are some concerns in the present manuscript so that major revision should be required before the judgment of acceptance.

1. Although the authors showed that Fn could enter ESCC cells, mechanisms by which live *Fusobacterium nucleatum* (Fn) or killed-Fn can impair T cells are unclear. Could Fn also enter T cells? How could killed-Fn impair T cells?
2. It would be helpful to focus Fn-Dps among various virulence factors of Fn.
3. When and how were clinical samples of ESCC collected before surgery or receiving PD-1 inhibitors, biopsy or resected specimens?
4. It would be helpful to explain why the authors use splenocyte and tumor cell coculture study.
5. The authors should discuss potential mechanisms by which Fn injection can promote liver and lung metastases in ESCC and anti-PD-L1 treatment cannot inhibit metastasis in mice underwent Fn infection.
6. The authors may remove a question mark in the Figure 6.

Point-by-point responses to the reviewers' comments

Reviewer 1:

1. The findings of this report agree with a previous one on colorectal cancer (Yaohui Gao et al, Signal Transduction and Targeted Therapy 2021) that found that Fn can induce PD-L1 expression. However, in the previous report high Fn levels were found to correlate with improved therapeutic responses to PD-1 blockade. The authors discuss this disagreement in the discussion, but not in sufficient depth.

Answer: Thank you for your question. We have added a related discussion in lines 353-357 and 363-368.

2. The authors demonstrate that Fn infection upregulates PD-L1 expression, however the immunotherapy used (Table S1) was anti-PD1. Should increase of PD-L1 have a dramatic effect?

Answer: Thank you for your question. Compared with anti-PD-1, the current clinical use of anti-PD-L1 is limited in ESCC; thus, the collected samples were all used with anti-PD-1. Usually, the expression of PD-L1 is used to select patients and analyze responses to both anti-PD-1 and anti-PD-L1 antibodies, although the basis for anti-PD-1 activity in PD-L1-negative tumors is incompletely understood.

In our study, as shown in Figure S3, the rate of PD-L1-positive cells did not differ between the R and NR groups. However, by combining the Fn abundance and PD-L1 expression for analysis, we found that a high protein level of PD-L1 and a high abundance of Fn DNA were more frequently found in the nonresponder (NR) group.

3. The study reveals a correlation between high levels of intratumoral Fn and failure in response to anti-PD-1 treatment in ESCC. This suggests that anti-PD-1 treatment may be futile in ESCC colonized by Fn. To resolve this point, it would be advantageous to conduct a TCGA dataset screening of all ESCC patients with Fn and compare the survival rates of patients who received anti-PD-1 treatment versus

those who did not. This analysis would offer valuable insights whether immunotherapy for patients with ESCC that is colonized by Fn is beneficial or not.

Answer: Thank you for your suggestion. We searched the TCGA dataset, and no data were available for the correlation between Fn abundance and anti-PD-1 efficacy in ESCC. In fact, Fn abundance and ICI efficacy have been reported in only a few publications. A recent study showed that a high abundance of Fn is correlated with decreased anti-PD-1 efficacy in colorectal cancer (*Jiang et al., Cell Host & Microbe. 2023;31:781-797*). A phase Ib/II clinical trial found that increased *Fusobacterium* is found in NR compared with R among metastatic colorectal cancer patients treated with a VEGF inhibitor plus anti-PD-1 (*Wang et al., Cell Reports Medicine. 2021;2:100383*). *Fusobacterium* is overrepresented in NR patients with locally advanced rectal cancer (*Yi et al., Clinical Cancer Research. 2021;27:1329-1340*). Another study found that the abundance of *Fusobacterium* is significantly increased in the NR group among patients with advanced solid tumors who were treated with anti-PD-1 and chemotherapy (*Wu et al., Frontiers in Oncology. 2022;12:887383*). Most of these studies performed the analyses at the “genus” level (*Fusobacterium*) and not at the “species” level. Until now, the interaction between Fn and immunotherapy and the underlying mechanism have not been clarified in cancer. Further study is needed in the future.

4. Use of an Fn mutant in Dps as a control, would greatly strengthen the conclusions of this study.

Answer: Thank you very much. We have attempted to construct an Fn-Dps mutant, but the Fn-Dps mutant cannot grow in isolation.

Some Dps mutants in aerobic bacteria can grow under aerobic cultivation conditions, but these are hypersensitive to H₂O₂. For example, when H₂O₂ is added at the beginning of the culture, *Dickeya dadantii*-Dps mutants are unable to grow (*Boughammoura et al., BioMetals. 2012;25:423-433*). Dps is important in conferring tolerance to hydrogen peroxide and for the survival of cells that enter the stationary phase of growth.

In microaerobic bacteria, the Dps mutant has difficulty surviving under microaerobic cultivation conditions. For example, wild-type *Helicobacter hepaticus* is able to survive in an atmosphere containing up to 6.0% O₂, the Dps mutant fails to grow in 3.0% or 6% O₂ and is also more sensitive to oxidative reagents such as H₂O₂ (Hong *et al.*, *Free Radic Res.* 2006;40:597-605).

Although it is strictly anaerobic to a certain extent, wild-type Fn is aerotolerant. The Fn-DPS mutant is hypersensitive to oxygen. We speculate that Fn-Dps is essential for *F. nucleatum* viability and survival and that the Fn-Dps mutant may not survive or be passaged, although we attempted to avoid aerobic operation as much as possible.

5. The report is concise and informative. However, in general, the description in the methods is short and in-sufficient. This is also true for the Figure legends, which makes following the presented results rather laborious.

Answer: Thank you very much for reminding us. We apologize for the simplicity of the “Methods” and “Figure legends” sections. We have added a detailed description in the revised manuscript.

6. In addition, insufficient details are supplied for the statistical analysis: “GraphPad Prism7 (GraphPad Software, CA, USA). The results are the means ± standard deviation (SD) of three independent experiments. 610 A p-value of less than 0.05 was considered statistically significant”. What were the tests used for each comparison? Where corrections made for multiple comparisons with the same data?

Answer: Thank you very much for reminding us. Before data analysis, the data were examined to determine whether they were normally distributed using the Shapiro–Wilk normality test. For normally distributed data, independent sample *t* tests were used for comparisons of two groups, and one-way ANOVA was used for comparisons of three or more groups. For data that were not normally distributed, the nonparametric Mann–Whitney test was used. We rewrote the “Statistical analysis” section in lines 670-684.

7. It is stated that “The results are the means \pm standard deviation (SD) of three independent experiments.” Please confirm that this (“of three independent experiments”) is true for each displayed result.

Answer: Thank you very much for reminding us. We apologize for this lapse and have corrected this sentence to “The results are presented as the means \pm SDs of three independent *in vitro* experiments at the cellular level” in lines 671-672.

8. Furthermore, Median rather than Mean seems more fitting for most of the displayed results that do not appear to be normally distributed.

Answer: Thank you for your suggestion. We used “Median” for the results in Figure 1A and S14B, and used “Mean” for the other results.

9. Add the “n” value, in the legend of each panel of each figure.

Answer: Thank you very much for reminding us. We have added the “n” values to the experiments related to clinical samples or animal samples. All cell experiments were independently repeated three times.

10. All bacteria names (including in references) should appear in “italics” style.

Answer: Thank you very much for reminding us. We have italicized all bacteria names in the revised paper accordingly.

11. figs 1 b-c: What is the cutoff separating CD274 high from CD274 low? Please present an additional corresponding color-coded (red, blue) graph with all of the points (values) of the CD274 high and low tumors.

Answer: Thank you for your question. According to the cutoff value of CD274, the samples were divided into high-CD274-expression or low-CD274-expression groups, and the cutoff value was generated by R programming language automatic analysis to minimize the *p* value. Figures were implemented in the R programming language. The “cutpoint” is shown in a separate Excel file named “Source Data”.

The “cutpoint” of CD274 for the OS analysis was 1.257, and that for the DFS analysis was 0.941.

12. Data verifying the specificity of the homemade antibodies used for identifying Fn in the tissues, and the antibodies used to detect Dps is not provided.

Answer: Thank you very much for reminding us. We have added the results regarding the validation of Fn and Fn-Dps antibodies specificity in Figure S16 in supplement.

13. In Fig S3 it is stated that “C57BL/6 mice (5 male per group) were implanted with 5×10^6 AKR cells BEFORE Fn infection (i.v., 107 CFU/mouse) three times.” In the methods and in Fig 2A it is stated that “C57BL/6 mice (5 male per group) were implanted with 5×10^6 AKR cells AFTER Fn infection (i.v., 107 647 CFU/mouse) three times.” Which is the correct description for the experiment?

Answer: Thank you very much for reminding us. We apologize for the incorrect description for Figure S4 (originally Figure S3) and have corrected the Figure S4 legend.

14. In general, the protocols lack details and hard to reproduce. For example, in lines 448-450: “For double immunofluorescence (IF) staining, the cells were fixed, permeabilized and blocked using 4 % paraformaldehyde (Beyotime Biotechnology, Shanghai, China) according to the manufacturer’s protocol.” I find it hard to follow which manufacturer is referred to?

Answer: Thank you for your proposal. We apologize for the simplicity of the “Methods” section. We have corrected the laps in lines 506-507 and added a detailed description throughout the “Methods” section.

15. Line 146: “(P < 0.05)” can be deleted.

Answer: Thank you for your proposal. We have deleted it accordingly.

16. Lines 96-97 and Fig. 1c: “Notably, Fn was detectable in both tumor tissues and adjacent normal tissues (Fig.1C).” I can’t find the detailed description how this was performed. Furthermore, why was Fn quantified in the RNA level (RT-qPCR) and not in the DNA level (qPCR) (as I understand was performed for Fig 1F), which is more reliable for bacterial quantification.

Answer: Thank you very much for reminding us. We have corrected “Normal” to “Adjacent Normal” in Figure 1C. Immunofluorescence staining of Fn was performed using both ESCC patient tumor and adjacent normal tissue sections, and the results showed that Fn was detectable in both tumor tissues and adjacent normal tissues. We apologize for the innocent misrepresentation in the manuscript. We quantified Fn at the DNA level through qPCR experiments and have corrected the information in the Figure 1D, 1K and 2C legends.

17. Please supply proof of the specificity of the primers use, demonstrate that a single band, with the expected size and sequence is obtained using each of these primers sets.

Answer: Thank you for your proposal. We determined the specificity of the primers by a BLAST search (<https://www.ncbi.nlm.nih.gov/tools/primer-blast/index.cgi>). The details are provided in Table S5 in the online supplement.

18. Fig. 1F: What do the blue and red colors represent, and what is “Fold Change”?

Answer: Thank you for the question. The tumor tissues are represented in red, and the paired adjacent normal tissues are represented in blue. We added a note to the Figure 1F legend. To determine the “fold change”, one sample was designated as the reference, and relative Fn or PD-L1 expression levels in other samples were calculated.

19. Fig. 1K: Is “Fold change” the correct title in the y axis?

Answer: Thank you for the question. To determine the “fold change”, one sample was designated as the reference, and relative Fn DNA levels in other samples were

calculated.

20. In Fig S6, is FITC representing the tumor cells (please indicate in the Fig legend)?

Please supply the statistical analysis for Fig S6b.

Answer: Thank you for the question. FITC represents the tumor cells, and we have amended the legend for Figure S8 (originally Figure S6). The statistical analysis of the data in Figure S8B is shown in Figure 4B.

21. Lines 164-165: rephrase: “Furthermore, Fn infection could not rescue the cytotoxic effect from splenocytes after treatment with the PD-1/PD-L1 blockade BMS202”.

Answer: Thank you for your proposal. We rephrased this sentence as follows (in lines 176-178): “Furthermore, Fn infection could not rescue the cytotoxic effect from splenocytes, even with the use of a PD-1/PD-L1 blockade BMS202.”

22. Lines 177-178: Where is the data supporting the conclusion that “Of note, live Fn culture from the infected cells revealed that Fn cannot survive inside T cells”? I would also restrict this to “under the tested conditions, live Fn culture from the infected cells revealed that Fn cannot survive inside T cells”. Please also show a control demonstrating that the fusobacteria were killed by the T cells, and not by the exposure to oxygen.

Answer: Thank you for the question. We performed an Fn intracellular survival assay of Jurkat cells and added the results to Figure 4J-K. Based on the results of intracellular survival assays (Figure 4J-K and Figure 5B-C), we found that the survival and multiplication of intracellular Fn could be maintained for 72 h in tumor cells, but completely opposite findings were obtained in Jurkat cells. The number of viable Fn decreased sharply in Jurkat cells after infection for 24 h to 72 h, and Jurkat cells exhibited obvious cell death after infection with Fn for 48 h (Figure 4H). Therefore, we hypothesize that Fn could induce apoptosis and inhibit the proliferation of T cells. After T-cell death, Fn is forced to transfer from the intracellular to extracellular environment; finally, as strictly anaerobic bacteria, Fn

cannot survive in a cell-free aerobic environment.

23. Lines 191-193 and Fig 4 B-C: is the increase in the numbers of intracellular fusobacteria statistically significant?

Answer: Thank you for the question. We have modified Figure 5B-C (originally Figure 4B-C) and their legends (lines 799-800). The numbers of viable Fn were significantly increased in E109 and Kyse150 cells in 72 h compared with 24 h, whereas no significant difference was found in AKR cells. However, the numbers of viable Fn in per ESCC cells (including three cell lines) were significantly increased at 72 h compared with 24 h, and we think that active proliferation of Fn could be obtained in three host ESCC cell lines.

24. Line 499: What is IP in “washed five times with IP and...”.

Answer: Thank you for the question. “IP” means cell lysis buffer for IP (immunological precipitation), and we have corrected this text in lines 555-556.

25. Lines 249-250, and legend to Fig 5N: I might have missed it, but I fail to find the discretion of the mutagenesis performed in the CD274 gene promoter region (including the description and the nature of the mutations).

Answer: Thank you for the question. To investigate whether CD274 is a direct target gene of ATF3, we predicted potential ATF3-binding sites in the human CD274 gene promoter region using the JASPAR database (<https://jaspar.genereg.net/>). As shown in Figure S14G, we found two predicted ATF3-binding motifs, and promoter constructs containing mutations in these two regions were generated to cause ATF3-binding deficiency. All experimental plasmids (M35 empty vector, M35-ATF3, PL01-CD274, PL01-CD274-ATF3 mutation vector) were synthesized by GeneCopoeia (MD, USA). The details are shown in lines 580-583 and Figure S14G legends.

26. Is the ATF3 binding motif in the CD274 promoter conserved both in human and in

mouse (explaining how Fn-Dps can operate in both species)?

Answer: Thank you for the question. We predicted ATF3 transcription factor-binding sites in the mouse CD274 promoter, and we found 8 putative sites (only 2 sites were predicted in the human CD274 promoter). Liu *et al* showed that ATF3 binds directly to the PD-L1 promoter in human melanoma cell lines and that shAtf3 also reduces the upregulation of PD-L1 expression in B16F10 xenografts (Liu *et al.*, *Cancer Cell*. 2020;37:324–339), and these results support our hypothesis and indicate that ATF3 can bind to the PD-L1 promoter in both humans and mice. Although the ATF3-binding motif in the CD274 promoter differs among humans and mice, it can be inferred from the binding site prediction results that the binding sites of ATF3 and PD-L1 are not unique. In brief, we think that ATF3 can directly regulate PD-L1 gene expression in both human and mouse cells.

27. Line 296: Define “NSCLC”.

Answer: Thank you very much for reminding us. NSCLC is the abbreviation for non-small cell lung cancer, and this abbreviation has been defined in line 299.

28. 312-314: Please rephrase “Liver metastasis could promote antigen-specific T-cell apoptosis and eventual immune therapy efficacy diminishment”.

Answer: Thank you for your proposal. We rephrased this sentence as follows (in lines 343-345): “Liver metastasis could promote antigen-specific T-cell apoptosis and an eventual attenuation in immune therapy efficacy.”

29. Line 316: Delete “mentioned”

Answer: Thank you for your proposal. We have deleted it accordingly.

30. 341-343: Rephrase “by binding to the PD-L1 gene promoter ATF3.”

Answer: Thank you for your proposal. We rephrased this sentence as follows (in lines 383-384): “by binding to the transcription factor ATF3.”

31. Carefully go over the English style in the “In summary” paragraph ending the discussion.

Answer: Thank you for your proposal. We have re-edited the “In summary” paragraph in lines 381-390.

Reviewer 3:

1. While the mouse findings are interesting, the findings would be strengthened if this could be repeated in a second ESCC line. Or by using the 4-NQO model if in need of an immune competent model

Answer: Thank you for your proposal. We repeated the experiment using E109 tumor-bearing NSG mice, and the results are shown in Figure 3 H-K.

2. The experimental design for the mouse experiments in Figure 2 seems inadequate. For In order to mimic patient treatments, it would have been better to let the tumor grow to a certain size and then to perform infections and PD-L1 treatment.

Answer: Thank you for your proposal. We have repeated the experiment. The mice were infected with Fn and administered a regular dose with α PD-L1 when the tumor volume reached 250~350 mm³, and the results are shown in Figure S6.

3. The use of certain ESCC cell lines in some experiments and other ones in different experiments is very confusing. It appears like the authors picked the cell line that would give them the result they wanted. Performing each experiments in at least 2 cell lines would be better

Answer: Thank you for your proposal. We have added one cell line to three cell level experiments: three-dimensional visualization of α -tubulin and Fn in CD3⁺T cells (Figure 4F), and two Co-IP experiments in E109 cells (Figure 6I and 6M). Other cell level experiments in our study were all performed using at least 2 cell lines.

4. For Figure 5, the rationale for ATF3 and the data presented is weak. The correlation between ATF3 and PD-L1 is low.

Answer: Thank you for the question. The relationship between ATF3 and PD-L1 was first reported by Liu *et al.* (Liu *et al.*, *Cancer Cell.* 2020;37:324–339). This study fully demonstrated that ATF3 is a major positive regulator of PD-L1 mRNA expression and that ATF3 binds directly to the PD-L1 promoter. Our study confirmed the effect of ATF3-PD-L1 binding using overexpressing and knockdown cells and Co-IP experiments with two types of ESCC cells. In addition, according to *GEPiA* (<http://gepia.cancer-pku.cn/>), ATF3 exhibits a significant positive correlation with PD-L1 expression in all types of cancer (as shown in the following figure, $p=0.0098$). We then validated the direct regulation and binding effect between Fn-Dps and ATF3 by using different experimental approaches (Co-IP, ChIP–qPCR, immunofluorescence colocalization and dual-luciferase reporter gene assays). Collectively, our results demonstrate that Fn-Dps could induce PD-L1 expression by upregulating its transcription through the activation of ATF3 in ESCC cells.

5. Important details in the method section are lacking for the patient cohort undergoing anti-PD-1 treatment. First of all, it appears that some patients underwent chemotherapy while other did not. Were these cofounders taken into consideration? Second of all, the criterias for patients to be classified as responders vs non-responders were not mentioned

Answer: Thank you for the question. Until now, immunotherapy combined with chemotherapy has been more clinically common in ESCC therapy. If the samples were collected exclusively from ESCC patients who were treated with anti-PD-1 therapy, we could only obtain a very limited number of samples. We rearranged our ELISA data (Figure 1G) and analyzed samples treated with anti-PD-1 alone. As shown in the figure below, the NR group (n=9) had higher serum anti-Fn IgG levels than the R group (containing 8 PR and 5 SD). This result is consistent with those shown in Figure 1G.

ESCC patients were defined based on Response Criteria in Solid Tumors (RECIST) 1.1. Based on magnetic resonance imaging (MRI) or computed tomography (CT) findings, their responses to treatment were evaluated as a partial response (PR), stable disease (SD), or progressive disease (PD). Clinical response (R) was defined as PR and SD, whereas PD was defined as nonclinical response (NR). We have added this information in lines 404-409 and Figure 1H.

6. This manuscript would really benefit from professional English editing services

Answer: Thank you for your proposal. The revised manuscript has been edited by professional editing service American Journal Experts (AJE), and we hope that the current version of manuscript reaches the standard for acceptance.

[REDACTED]

7. In the results, mention that analyzed Fn in ESCA tissue? Is that a typo or was it a combination of ESCC and EAC? This should be clarified

Answer: Thank you for the question. ESCA has two main subtypes, ESCC and EAC. ESCC accounts for approximately 90% of cases of ESCA worldwide (Rustgi et al., N Engl J Med. 2014;371(26):2499-509), and we selected ESCC for our study. We used ESCC cell lines (E109 and Kyse150) and collected ESCC patient samples in our study. Since the TCGA database only has ESCA options and is not subdivided into ESCC or EAC, the TCGA data analyses in the figures (Figure 1A-1B, Figure 6G and Figure S14B-C) were performed using ESCA data.

8. Figure 1 A, C: normal should be labeled adjacent normal. Same for methods and results. This is misleading to call them normal.

Answer: Thank you very much for reminding us. We have corrected this information in our revised manuscript and supplemental materials accordingly.

9. Figure 1D: says n=12 but in the text the authors mentions that they collected 6 specimen. Where is the n=12 coming from? Counting 6 ESCC and 6 adjacent normal is wrong and misleading

Answer: Thank you very much for reminding us. We collected 12 ESCC tissue samples and paired adjacent normal tissue samples. The volume of 6 pairs was too small for the extraction of both protein and DNA; thus, we extracted DNA from 12 tumor tissues and paired adjacent normal tissues and only extracted protein from 6 of the 12 tissue pairs. Therefore, “n=12” in Figure 1D is correct. We have corrected the number in line 394 in the “Methods” section.

10. Figure 1E, given that GAPDH is uneven, densitometry should be performed. Same for all other WB included.

Answer: Thank you for your proposal. We have added the results from densitometry analyses to Figure 1E, Figure 5K, Figure 6H (the result is shown in Figure S14D) and Figure 6J (the result is shown in Figure S14F).

11. How do the authors reconcile that they observed no change in metastasis in the xenograft model, but that they did in the tail vein model?

Answer: Thank you for the question. As mentioned in lines 129-130, we found significant liver metastases and lung inflammation in the xenograft model. Notably, the length of the two animal experiments was different: 18 days in the xenograft model and 35 days in the tail vein model. Tail vein injection of tumor cells is more likely to cause lung and liver metastasis than subcutaneous injection. Combining the above two points, we observed more obvious tumor metastasis in the tail vein model than in the xenograft model.

12. Figure 2D should say volume not volumn

Answer: Thank you very much for reminding us. We apologize for this mistake caused by our carelessness, and we have corrected this mistake accordingly.

13. To strengthen the rationale for the experiments in Figure 3, the authors should show a staining of a positive Fn and CD3 co-staining in ESCC

Answer: Thank you for your proposal. We have added the results in Figure 3G.

14. Figure 4L does not show that much expression of Fn. Looking at Figure, it appears that Fn levels are only going up 10 folds compared to the 4000 fold in human. ESCC. Furthermore, quantitative experiments of Fn and PD-L1, such as WB and WB would be more convincing.

Answer: Thank you for the question. Figure 5L (originally Figure 4L) shows the results of IF staining of Fn in tumor tissues from C57BL/6 xenografts. Fn has been confirmed to be an intratumor bacterial species in some types of human tumor tissues (*Nejman et al., Science. 2020;368: 973–980*) but is not present in mouse tumor tissues. In our study, Fn was administered to tumor-bearing mice *via* tail vein injection. Therefore, it may not be feasible to compare Fn expression between human and mouse tumor tissues.

We detected the expression of Fn by qPCR and PD-L1 by WB. In general, there are no recognized specific protein markers to level Fn in WB experiments; thus, qPCR is better for the quantification of Fn. Several studies have also used qPCR to detect Fn (*Jiang et al., Cell Host & Microbe. 2023;31:781–797; Gao et al., Signal Transduct Target Ther. 2021;6(1):398; Chen et al., Gut Microbes. 2020;11(3):511–525*).

Reviewer 4:

1. Although the authors showed that Fn could enter ESCC cells, mechanisms by which live *Fusobacterium nucleatum* (Fn) or killed-Fn can impair T cells are unclear. Could Fn also enter T cells? How could killed-Fn impair T cells?

Answer: Thank you for the question. As shown in Figure 4F, Fn could enter CD3⁺T and Jurkat cells after infection for 8 h. However, the survival and multiplication of intracellular Fn could not be maintained in Jurkat cells after infection for up to 72 h, and the number of viable Fn showed a sharp decreasing trend (Figure 4J). Our findings showed that only live Fn could induce apoptosis, and both live Fn and killed-Fn inhibited cytokine secretion by T cells. Our previous study showed that Fn impairs the cytolytic function of peripheral blood lymphocytes by upregulating indoleamine 2,3-dioxygenase expression in host cells, leading to tryptophan

deficiency in the environment. Tryptophan is needed for T lymphocyte effector functions.

In addition, several studies have shown that virulence factors of Fn impairs lymphocytes or T cells functions, including *F. nucleatum* immunosuppressive protein (FIP) and lectin Fap2 (Demuth et al., *Infect Immun.* 1996;64:1335-41) (Parhi et al., *Nature Communications.* 2020;11:3259). FIP inhibits T-cell activation by impairing the expression of the proliferating-cell nuclear antigen of T-cells. We speculate that the reason why Fn impairs T cells may be related to some virulence factors, including FIP, Fap2 and other unexplored proteins, but further research is needed in the future.

The reason of killed-Fn impaired T cells may related to some virulence factors including proteins or polysaccharides. Lipopolysaccharide (LPS) can significantly inhibit the proliferation of T cells and suppress the functions of T cells (Sueyoshi et al., *J Biol Chem.* 2019;294:6283-6293). The LPS structure of Fn is different from that of other Gram-negative bacteria, and Fn-LPS may be potently cytotoxic to T cells (Vinogradov et al., *Carbohydr Res.* 2017;440-441:10-15) (Garcia-Vello et al., *Chembiochem.* 2021;22:1252-1260). It is possible that heat-killed Fn impair the function of T-cells by its LPS or unexplored proteins, but further research is needed in the future.

2. It would be helpful to focus Fn-Dps among various virulence factors of Fn.

Answer: Thank you very much for reminding us. We have added a related discussion in lines 292-296.

3. When and how were clinical samples of ESCC collected before surgery or receiving PD-1 inhibitors, biopsy or resected specimens?

Answer: Thank you for the question. We have added detailed information about the collected ESCC clinical samples in lines 399-404.

4. It would be helpful to explain why the authors use splenocyte and tumor cell

coculture study.

Answer: Thank you for your proposal. We have added the explanation in the lines 169-172.

5. The authors should discuss potential mechanisms by which Fn injection can promote liver and lung metastases in ESCC and anti-PD-L1 treatment cannot inhibit metastasis in mice underwent Fn infection.

Answer: Thank you for your proposal. We have added a related explanation in lines 325-333.

6. The authors may remove a question mark in the Figure 6.

Answer: Thank you for your proposal. We have removed the question mark in Figure 7 (originally Figure 6).

REVIEWERS' COMMENTS

Reviewer #1 (Remarks to the Author):

I appreciate the effort invested by the authors in addressing my comments. However, I believe there are still three key points that can be further improved:

1. Previous comment #4, "Use of an Fn mutant in Dps as a control, would greatly strengthen the conclusions of this study": Please refer to this comment also in the text, so that the readers will know that generation of the Dps mutant proved impossible.
2. Regarding previous comment #17, "Please supply proof of the specificity of the primers ...": For the primers used to quantify *F. nucleatum*, I fear that the "BLAST search" performed by the authors is not a sufficient testimony for specificity. Please demonstrate that a single band, with the expected size and sequence is obtained using F:5'-AAGCGCGTCTAGGTGGTTATGT-3; and R:5'-TGTAGTCCGCTTACCTCTCCAG-3' primers. This demonstration of specificity is important, as readers of this article might want to use these primers for future quantification of *F. nucleatum*.
3. Please refer to comment previous #26: "Is the ATF3 binding motif in the CD274 promoter conserved both in human and in mouse (explaining how Fn-Dps can operate in both species)?" also in the text, to convey your hypotheses to the reader.

Reviewer #3 (Remarks to the Author):

The authors have addressed the concerns from this reviewer adequately

Reviewer #4 (Remarks to the Author):

The manuscript is well revised according to reviewers' comments.

Point-by-point responses to the reviewers' comments

Reviewer 1:

1. Previous comment #4, "Use of an Fn mutant in Dps as a control, would greatly strengthen the conclusions of this study": Please refer to this comment also in the text, so that the readers will know that generation of the Dps mutant proved impossible.

Answer: Thank you for your suggestion. We have added a related discussion in lines 295-298.

2. Regarding previous comment #17, "Please supply proof of the specificity of the primers ...": For the primers used to quantify *F. nucleatum*, I fear that the "BLAST search" performed by the authors is not a sufficient testimony for specificity. Please demonstrate that a single band, with the expected size and sequence is obtained using F:5'-AAGCGCGTCTAGGTGGTTATGT-3; and R:5'-TGTAGTTCCGCTTACCTCTCCAG-3' primers. This demonstration of specificity is important, as readers of this article might want to use these primers for future quantification of *F. nucleatum*.

Answer: Thank you for your question. The results of melting curve and agarose gel analysis (as shown in Figure S17 in supplement file) showed that the Fn-primers are working well with satisfying specificity.

3. Please refer to comment previous #26: "Is the ATF3 binding motif in the CD274 promoter conserved both in human and in mouse (explaining how Fn-Dps can operate in both species)?" also in the text, to convey your hypotheses to the reader.

Answer: Thank you for your suggestion. We have added a related discussion in lines 302-307.